# Learning $U$-Statistics with Active Inference

Xiaoning Wang [* 1]   Yuyang Huo [* 1]   Liuhua Peng [2]   Changliang Zou [1]

## Abstract

$U$-statistics play a central role in statistical inference. In many modern applications, however, acquiring the labels required for $U$-statistics is costly. Motivated by recent advances in active inference, we develop an active inference framework for $U$-statistics that selectively queries informative labels to improve estimation efficiency under a fixed labeling budget, while preserving valid statistical inference. Our approach is built on the augmented inverse probability weighting $U$-statistic, which is designed to incorporate the sampling rule and machine learning predictions. We characterize the optimal sampling rule that minimizes its variance and design practical sampling strategies. We further extend the framework to $U$-statistic-based empirical risk minimization. Experiments on real datasets demonstrate substantial gains in estimation efficiency over baseline methods, while maintaining target coverage.

## 1. Introduction

$U$-statistics (Hoeffding, 1948) play a fundamental role in both classical and modern statistical learning and inference by providing a unified framework for estimating and making inferences on parameters that can be expressed as the expectation of a symmetric function of multiple random variables. Formally, let $\mathcal{Y}_n = \{Y_1, \ldots, Y_n\}$ denote i.i.d. samples drawn from a common distribution $\mathbb{F}_Y$, the parameter of interest is

$$\theta^* = \mathbb{E}\left[h\left(Y_1, \ldots, Y_r\right)\right],$$

where $h(\cdot)$ is a known kernel function symmetric in its $r$ arguments with $\mathbb{E}[h\left(Y_1, \ldots, Y_r\right)]^2 < \infty$. Given the sample

$\mathcal{Y}_n$, we can estimate $\theta^*$ using a $U$-statistic:

$$U\left(\mathcal{Y}_n\right) = \frac{1}{\binom{n}{r}} \sum_{\mathcal{C}_{n,r}} h\left(Y_{i_1}, \ldots, Y_{i_r}\right), \qquad (1)$$

where $\mathcal{C}_{n,r} = \{(i_1, \ldots, i_r) : 1 \leq i_1 < \cdots < i_r \leq n\}$ is the collection of all $r$-tuples of distinct indices.

$U$-statistics are unbiased and statistically efficient estimators that can capture pairwise and higher-order interaction structure, making them fundamental to a wide range of statistical problems. Representative examples include the Gini index (Yitzhaki, 2003) for measuring imbalance; the area under the ROC curve (AUC) (DeLong et al., 1988) for evaluating classification performance, which is equivalent to the Mann-Whitney $U$-statistic (Mann & Whitney, 1947); and Kendall's $\tau$ coefficient (Kendall, 1948) for measuring dependence. Beyond classical inference, $U$-statistics also arise in modern machine learning tasks, including ranking (Clémençon et al., 2008), representation learning (Saunshi et al., 2019) and metric learning (Jin et al., 2009). In this paper, we focus on non-degenerate $U$-statistics (Serfling, 2009), which admit asymptotic normality for inference.

Despite their broad applicability, learning $U$-statistics in practice is often hindered by limited labeled data. In many modern applications, the target variable $Y \sim \mathbb{F}_Y$ is unobserved, and only covariates $X \sim \mathbb{F}_X$ are available. Acquiring labels typically requires costly human annotation or experimental intervention. We therefore consider a setting with a fixed labeling budget $n_b \ll n$, under which labels can be queried for only a small subset of samples.

When covariates $X_1, \ldots, X_n$ drawn from $\mathbb{F}_X$ are available, a natural alternative is to leverage a machine learning model $\mu(X)$, which predicts the unknown label $Y$. Let $\hat{Y}_i = \mu(X_i)$, $i \in [n]$ be the predictions. A naive approach is to directly substitute these predictions for the true labels when computing the target $U$-statistic. Such plug-in strategies are common in medical diagnostics (Bhavsar et al., 2021), social science (Grimmer et al., 2021), and financial decision-making (Gu et al., 2020), where labels are expensive or delayed. However, such plug-in estimators are generally biased and can exhibit uncontrolled variance, particularly when the predictive model is misspecified or inaccurate.

Recently, active inference (Zrnic & Candès, 2024a) has emerged as a principled framework for improving estima-

[1] School of Statistics and Data Sciences, LPMC, KLM-DASR and LEBPS, Nankai University, Tianjin, China [2] School of Mathematics and Statistics, The University of Melbourne, Melbourne, Australia . Correspondence to: Liuhua Peng <liuhua.peng@unimelb.edu.au>, Changliang Zou <zoucl@nankai.edu.cn>.

*Proceedings of the 43$^{rd}$ International Conference on Machine Learning*, Seoul, South Korea. PMLR 306, 2026. Copyright 2026 by the author(s).

tion efficiency with valid statistical inference under limited labeling budgets by leveraging predictive models and adaptively querying labels that are expected to be most informative. Active inference can be viewed as an extension of active learning from prediction to inference (Ren et al., 2021). Existing works (Zrnic & Candès, 2024a; Li et al., 2025; Chen et al., 2025a) have primarily focused on convex M-estimation problems. These methods, however, are not directly applicable to $U$-statistics, whose combinatorial structure and dependence across sample tuples pose additional analytical and algorithmic challenges.

In this paper, we study learning $U$-statistics with active inference. Our goal is to design label-acquisition strategies that strategically select samples for labeling to minimize the estimation error of a target $U$-statistic, while enabling valid follow-up inference under a fixed labeling budget. Our contributions are as follows:

- Building on the proposed augmented inverse probability weighted $U$-statistic, we characterize the optimal sampling rule and develop an active sampling strategy beyond existing rules (Zrnic & Candès, 2024a).
- We develop a unified theory for active $U$-statistics, establishing asymptotic normality and enabling computable confidence intervals. A coupling-based proof handles the dependence introduced by data-adaptive sampling policies.
- We extend the proposed active inference framework to $U$-statistic–based empirical risk minimization via a two-stage procedure, with optimal sampling policy and rigorous theoretical guarantees. This formulation encompasses a wide range of modern machine learning tasks, including ranking.
- Numerical experiments on real datasets demonstrate that our method achieves substantially improved estimation efficiency compared to existing baselines.

## 1.1. Related works

$U$-**statistics**  $U$-statistics have been extensively studied as a general framework for estimating functionals defined by pairwise or higher-order kernels. Foundational work established their unbiasedness and asymptotic behavior (Hoeffding, 1948; Serfling, 2009; Peel et al., 2010), while more recent research has focused on scalability, inference, and learning-theoretic properties. A major line of research addresses the computational challenges of $U$-statistics, including distributed implementations (Chen & Peng, 2021) and incomplete or randomized $U$-statistics (Chen & Kato, 2019). Beyond computation, further extensions consider inference for degenerate $U$-statistics (Chen, 2018) and estimation under privacy constraints (Chaudhuri et al., 2024).

Most closely related to our work is the study on semi-supervised $U$-statistics (Kim et al., 2025), which leverages predictive models to incorporate unlabeled data into $U$-statistics. In contrast, we study an active inference setting in which labeled data are not pre-given but are adaptively acquired under a fixed labeling budget. The method of Kim et al. (2025) can be viewed as a special case of our framework corresponding to uniform sampling. By explicitly designing data-driven sampling policies, our approach is better suited to label-budget–constrained scenarios and enables variance-optimal estimation with valid inference.

**Active inference**  Active data acquisition has a long history in statistics and machine learning. Classical optimal experimental design studies the selection of design points to optimize information-based criteria (Ma et al., 2015; Wang et al., 2018; Chen et al., 2025b). In machine learning, active learning focuses on selecting informative unlabeled instances to query in order to improve predictive accuracy (Settles, 2012; Shui et al., 2020; Min et al., 2025).

More recently, *active statistical inference* has emerged as a distinct paradigm, where the objective is not prediction but valid inference under a limited labeling budget. This line of work is closely related to semi-supervised estimation (Zhang et al., 2019; Zhu et al., 2023; Wen et al., 2025) and prediction-powered inference (Angelopoulos et al., 2023a; Zrnic & Candès, 2024b; Chatzi et al., 2024; Kluger et al., 2025), which leverage auxiliary predictive models to improve efficiency. Subsequent advances have addressed label-efficient model evaluation (Angelopoulos et al., 2025), LLM-annotation-based conclusion (Gligorić et al., 2025), robust sampling (Li et al., 2025), and balanced or variance-aware sampling (Chen et al., 2025a). While existing methods primarily target general M-estimation problems, $U$-statistics pose additional challenges due to their pairwise or higher-order structure, which is the focus of this work.

## 2. Active Inference for $U$-statistics

### 2.1. AIPW $U$-statistic

Suppose $\{(X_i, Y_i)\}_{i=1}^n$ are i.i.d.. We observe the covariates $X_1, \ldots, X_n$, drawn from $\mathbb{F}_X$ supported on $\mathcal{X}$, and employ a sampling rule $\pi : \mathcal{X} \to [0, 1]$. For each $i \in [n]$, the label $Y_i$ is collected with probability $\pi_i = \pi(X_i)$. Let $\xi_i \sim$ Ber $(\pi(X_i))$ denote the indicator that $Y_i$ is collected or not. The total number of acquired labels is $n_{\text{lab}} = \sum_{i=1}^n \xi_i$. To satisfy the labeling budget constraint, $\pi(\cdot)$ will be rescaled such that $\mathbb{E}(n_{\text{lab}}) = \sum_{i=1}^n \mathbb{E}[\pi(X_i)] = n\mathbb{E}[\pi(X)] \leq n_b$.

Since the sampling rule $\pi(\cdot)$ induces selection bias, $U(\{Y_i : \xi_i = 1\})$ as in (1) is biased for $\theta^*$. We therefore consider the inverse probability weighting (IPW) $U$-statistic (Horvitz & Thompson, 1952):

$$U_{\text{IPW}}^\pi(\mathcal{Y}_n) = \frac{1}{\binom{n}{r}} \sum_{\mathcal{C}_{n,r}} h(Y_{i_1}, \ldots, Y_{i_r}) \frac{\xi_{i_1} \cdots \xi_{i_r}}{\pi_{i_1} \cdots \pi_{i_r}}. \quad (2)$$

This estimator relies only on the observed labels through the indicators $\{\xi_i\}_{i=1}^n$, and unlabeled observations contribute nothing beyond determining the sample probabilities. To additionally exploit the unlabeled covariates, we introduce an augmented inverse probability weighting (AIPW) $U$-statistic (Cassel et al., 1976) that incorporates machine learning predictions $\hat{\mathcal{Y}}_n = \{\hat{Y}_i = \mu(X_i)\}_{i=1}^n$. Regarding the fact that the plug-in estimator $U(\hat{\mathcal{Y}}_n)$ is biased, we correct its bias using an IPW term and define the AIPW $U$-statistic as

$$
\begin{aligned}
U_{\mathrm{AIPW}}^\pi = {} & \frac{1}{\binom{n}{r}} \sum_{\mathcal{C}_{n,r}} h\big(\hat{Y}_{i_1}, \ldots, \hat{Y}_{i_r}\big) \\
& + \frac{1}{\binom{n}{r}} \sum_{\mathcal{C}_{n,r}} \Delta_h(i_1, \ldots, i_r) \frac{\xi_{i_1} \cdots \xi_{i_r}}{\pi_{i_1} \cdots \pi_{i_r}}
\end{aligned}
\tag{3}
$$

with $\Delta_h(i_1, \ldots, i_r) = h\left(Y_{i_1}, \ldots, Y_{i_r}\right) - h\big(\hat{Y}_{i_1}, \ldots, \hat{Y}_{i_r}\big)$.

**Proposition 2.1** (Unbiasedness of the AIPW $U$-statistic). *The AIPW $U$-statistic $U_{\mathrm{AIPW}}^\pi$ satisfies $\mathbb{E}[U_{\mathrm{AIPW}}^\pi] = \theta^*$.*

A natural question arises: how should one choose the sampling rule $\pi(\cdot)$ to minimize the variance of the AIPW $U$-statistic. In Zrnic & Candès (2024a), the optimal sampling design for estimating a mean-type functional yields $\pi(X_i) \propto \sqrt{\mathbb{E}[|\hat{Y}_i - Y_i|^2 \mid X_i]}$, so that sampling concentrates on points with large residual uncertainty. However, this optimality does not hold for $U$-statistics.

## 2.2. Optimal sampling rule for AIPW $U$-statistics

To derive the optimal sampling rule, we analyze the variance of $U_{\mathrm{AIPW}}^\pi$ via the Hoeffding decomposition. First, we write

$$
U_{\mathrm{IPW}}^\pi (\mathcal{Y}_n) = \theta^* + \frac{r}{n} \sum_{i=1}^n \left[ h_1(Y_i) \frac{\xi_i}{\pi_i} - \theta^* \right] + R_{\mathrm{IPW}}^\pi (\mathcal{Y}_n),
$$

where $h_1(y) = \mathbb{E}[h(Y_1, \ldots, Y_r) \mid Y_1 = y]$ is the first-order projection, and $R_{\mathrm{IPW}}^\pi (\mathcal{Y}_n)$ is the remainder. Here we assume $\mathrm{Var}(h_1(Y)) > 0$, so that the $U$-statistics is non-degenerate. Similarly, for the IPW $U$-statistic based on $\hat{\mathcal{Y}}_n$, we have

$$
U_{\mathrm{IPW}}^\pi(\hat{\mathcal{Y}}_n) = \theta^\mu + \frac{r}{n} \sum_{i=1}^n \left[ h_1^\mu(\hat{Y}_i) \frac{\xi_i}{\pi_i} - \theta^\mu \right] + R_{\mathrm{IPW}}^\pi(\hat{\mathcal{Y}}_n),
$$

where $h_1^\mu(y) = \mathbb{E}[h(\hat{Y}_1, \ldots, \hat{Y}_r) \mid \hat{Y}_1 = y]$ with the assumption that $\mathbb{E}[h(\hat{Y}_1, \ldots, \hat{Y}_r)]^2 < \infty$ and $\mathrm{Var}(h_1^\mu(\hat{Y})) > 0$ for the non-degeneracy of $U_{\mathrm{IPW}}^\pi(\hat{\mathcal{Y}}_n)$, $\theta^\mu = \mathbb{E}[h(\hat{Y}_1, \ldots, \hat{Y}_r)]$, and $R_{\mathrm{IPW}}^\pi(\hat{\mathcal{Y}}_n)$ is the corresponding remainder term. Recall that $U_{\mathrm{AIPW}}^\pi = U(\hat{\mathcal{Y}}_n) + U_{\mathrm{IPW}}^\pi(\mathcal{Y}_n) - U_{\mathrm{IPW}}^\pi(\hat{\mathcal{Y}}_n)$, where $U(\hat{\mathcal{Y}}_n) \approx \theta^\mu + rn^{-1} \sum_{i=1}^n \left[ h_1^\mu(\hat{Y}_i) - \theta^\mu \right]$. Combining the

above decompositions yields

$$
\begin{aligned}
U_{\mathrm{AIPW}}^\pi = {} & \theta^\mu + \frac{r}{n} \sum_{i=1}^n \left[ h_1^\mu(\hat{Y}_i) - \theta^\mu \right] + (\theta^* - \theta^\mu) \\
& + \frac{r}{n} \sum_{i=1}^n \left\{ \left[ h_1(Y_i) - h_1^\mu(\hat{Y}_i) \right] \frac{\xi_i}{\pi_i} - (\theta^* - \theta^\mu) \right\} + R,
\end{aligned}
\tag{4}
$$

where $R$ collects remainder terms of small orders. Dropping the negligible remainder, a direct calculation gives

$$
\begin{aligned}
\mathrm{Var}(U_{\mathrm{AIPW}}^\pi) \approx {} & \frac{r^2}{n} \mathrm{Var}(h_1(Y)) \\
& + \frac{r^2}{n^2} \sum_{i=1}^n \mathbb{E}\left\{ \mathbb{E}\{[h_1(Y_i) - h_1^\mu(\hat{Y}_i)]^2 \mid X_i\} \cdot \left( \frac{1}{\pi_i} - 1 \right) \right\},
\end{aligned}
\tag{5}
$$

which motivates the optimal sampling rule in next Theorem.

**Theorem 2.2** (Optimal sampling probabilities for the AIPW $U$-statistic). *Fix a labeling budget $n_b$. Let $s(X) := \mathbb{E}[|h_1(Y) - h_1^\mu(\hat{Y})|^2 \mid X]$ and set*

$$
\pi^*(X) = \min\left\{ 1, \; \frac{n_b}{n} \cdot \frac{\sqrt{s(X)}}{\mathbb{E}[\sqrt{s(X)}]} \right\}.
$$

*Among all sampling rules $\pi : \mathcal{X} \to [0, 1]$ with $\mathbb{E}[\pi(X)] \leq n_b/n$, we have*

$$
\mathrm{Var}(U_{\mathrm{AIPW}}^\pi) \; \geq \; \mathrm{Var}\left( U_{\mathrm{AIPW}}^{\pi^*} \right), \qquad \text{as } n \to \infty.
$$

Theorem 2.2 indicates that the optimal sampling probability should satisfy $\pi(X) \propto \sqrt{\mathbb{E}[|h_1(Y) - h_1^\mu(\hat{Y})|^2 \mid X]}$ to minimize the asymptotic variance of $U_{\mathrm{AIPW}}^\pi$ under the budget constraint. In particular, the optimal rule is driven by the residual uncertainty in the first-order Hoeffding projection, $|h_1(Y) - h_1^\mu(\hat{Y})|$, rather than the prediction residual $|\hat{Y} - Y|$ that is natural for mean estimation (Zrnic & Candès, 2024a). As a sanity check, when $r = 1$ and $h(y) = y$, our rule reduces to the classic one based on $|\hat{Y} - Y|$. Appendix C.2 provides a numerical comparison illustrating that sampling based on $|\hat{Y} - Y|$ can be suboptimal for $U$-statistic targets.

**Normalized AIPW $U$-statistics** A key difference between the AIPW $U$-statistic $U_{\mathrm{AIPW}}^\pi$ in (3) and its classical i.i.d. counterpart $U(\mathcal{Y}_n)$ in (1) is that it involves products of inverse inclusion probabilities. Although $U_{\mathrm{AIPW}}^\pi$ is asymptotically valid, its multiplicative inverse-probability structure typically leads to severe finite-sample variance inflation, a phenomenon well known in survey sampling.

To address this, we extend the idea of Hájek normalization (Trotter & Tukey, 1956): instead of the fixed combinatorial

normalizer $\binom{n}{r}$, we use a random, data-adaptive estimate:

$$\widehat{N} = \sum_{\mathcal{C}_{n,r}} \xi_{i_1} \cdots \xi_{i_r} / \pi_{i_1} \cdots \pi_{i_r}. \tag{6}$$

Since $\mathbb{E}[\widehat{N}] = \binom{n}{r}$, $\widehat{N}$ is an unbiased estimator of the total number of $r$-tuples and therefore provides a natural random normalization. Hájek-type normalization is widely used in survey sampling to stabilize inverse-probability-weighted estimators by reducing the impact of extreme inverse-probability weights. See discussions in Särndal et al. (2003) and Datta & Polson (2022). Our contribution here is to generalize this normalization principle from mean estimation to the $U$-statistic setting.

We accordingly define the normalized AIPW $U$-statistic as

$$U_{\text{AIPW}}^{\pi,N} = \frac{1}{\binom{n}{r}} \sum_{\mathcal{C}_{n,r}} h(\hat{Y}_{i_1}, \ldots, \hat{Y}_{i_r})$$
$$+ \frac{1}{\widehat{N}} \sum_{\mathcal{C}_{n,r}} \Delta_h(i_1, \ldots, i_r) \frac{\xi_{i_1} \cdots \xi_{i_r}}{\pi_{i_1} \cdots \pi_{i_r}}. \tag{7}$$

Theorem 2.3 shows that $U_{\text{AIPW}}^{\pi,N}$ is asymptotically unbiased for $\theta^*$. Nevertheless, consistent with the previous survey-sampling literature, our empirical results show that this normalization can significantly reduce the variance of $U_{\text{AIPW}}^{\pi}$ in practice. See Appendix C.4 for a direct comparison.

**Theorem 2.3** (Asymptotic unbiasedness of the normalized AIPW $U$-statistic). *The normalized AIPW $U$-statistic $U_{\text{AIPW}}^{\pi,N}$ is asymptotically unbiased for $\theta^*$, i.e., $\mathbb{E}[U_{\text{AIPW}}^{\pi,N}] = \theta^* + o(1)$.*

**Trimming** The normalized AIPW $U$-statistic can still suffer from high variance when some $\pi_i$ are very small. To further control variance, we adopt an adaptive trimming strategy. Specifically, define $\pi_i^{\text{trim}} = \tau \pi_i + (1 - \tau) \pi^{\text{unif}}$, where $\tau \in [0, 1]$ is a weighting parameter and $\pi^{\text{unif}} = n_b/n$ is the uniform sampling policy. This ensures that the minimal sampling probability is bounded below by $(1 - \tau) n_b/n$, thereby preventing excessive variance inflation due to rare sampling events. The tuning parameter $\tau$ is fixed at 0.7 throughout our experiments. It can be also selected via the robust sampling strategies in Li et al. (2025).

### 2.3. Active $U$-statistics and variance estimation

We now describe the construction of the active sampling rule and the resulting active $U$-statistic used for inference. Assume we have access to a pre-trained prediction model $\mu(\cdot)$, together with a small amount of auxiliary labeled data obtained, for example, through an initial uniform sampling stage. This setup is standard in active statistical inference (Zrnic & Candès, 2024a; Li et al., 2025), where predictive models are used to guide subsequent sampling strategies.

Recall that the optimal sampling rule depends on $s(X) = \mathbb{E}[|h_1(Y) - h_1^\mu(\hat{Y})|^2 \mid X]$, we need to estimate the unknown first-order projections $h_1$ and $h_1^\mu$ to determine the sampling probabilities. They can be accurately approximated using only a limited number of queried labels. Let $\hat{h}_1$ the $\hat{h}_1^\mu$ denote estimators of $h_1$ and $h_1^\mu$, respectively. We then learn a regression function

$$V(x) = \mathbb{E}[|\hat{h}_1(Y) - \hat{h}_1^\mu(\hat{Y})| \mid X = x],$$

which approximates $\sqrt{s(x)}$ and serves as a data-driven proxy for the optimal sampling score.

Given the estimated score $V(X_i)$, we define the initial active sampling probabilities by $\hat{\pi}(X_i) = n_b V(X_i)/\sum_{j=1}^n V(X_j)$ so that $\sum_{i=1}^n \hat{\pi}(X_i) = n_b$. To prevent extremely small probabilities, we apply trimming: $\hat{\pi}_\tau(X_i) = \tau \hat{\pi}(X_i) + (1 - \tau) n_b/n$. Finally, the active $U$-statistic is defined as the normalized AIPW $U$-statistic $U_{\text{AIPW}}^{\hat{\pi}_\tau,N}$ in (7), evaluated using the trimmed active sampling probabilities $\hat{\pi}_\tau(X_i)$. The implementation and computation of the active $U$-statistic is summarized in Algorithm 1. To ensure valid inference and avoid overfitting, we employ sample splitting for nuisance parameter estimation; see details in Appendix B.1.

We briefly discuss the computational cost of the active $U$-statistic. The dominant cost comes from the plug-in $U$-statistic computed over all predicted labels, which scales as $O(n^r)$. In contrast, the correction term based on queried true labels only involves the labeled subset and scales as $O(n_b^r)$. The additional cost of estimating $V(x)$ and constructing $\hat{\pi}_\tau$ is based on a small pilot labeled set and is typically much smaller. Detailed running-time comparisons are provided in Appendix B.2.

---

**Algorithm 1** Active $U$-statistics

**Require:** Unlabeled data $X_1, \ldots, X_n$; Budget $n_b$; Pre-trained predictive model $\mu(\cdot)$; kernel function $h$ and degree $r$; weight parameter $\tau$.

1: Calculate $\hat{Y}_i = \mu(X_i)$, $i \in [n]$.
2: Obtain uncertainty measure $V(x)$ based on $\mu(\cdot)$.
3: Calculate initial active sampling probabilities: $\hat{\pi}(X_i) = n_b V(X_i)/\{\sum_{j=1}^n V(X_j)\}$, $i \in [n]$.
4: Trimming: $\hat{\pi}_\tau(X_i) = \tau \hat{\pi}(X_i) + (1 - \tau) n_b/n$, $i \in [n]$.
5: Sample labeling decisions according to $\hat{\pi}_\tau(X_i)$: $\xi_i \sim \text{Bern}(\hat{\pi}_\tau(X_i))$, $i \in [n]$. Collect labels $\{Y_i : \xi_i = 1\}$.
6: Compute $\widehat{N}$ in (6) and the normalized AIPW $U$-statistic via (7) using $\hat{\pi}_\tau(X_i)$.

---

**Asymptotic normality** Now we establish the asymptotic normality of the proposed active $U$-statistics.

**Theorem 2.4** (CLT for active $U$-statistics). *Let*

$$\pi_\tau^*(X_i) = \frac{n_b}{n}\left(\tau\frac{V(X_i)}{\mathbb{E}[V(X_1)]} + 1 - \tau\right).$$

*Under Assumption A.1 in Appendix A.1, as $n \to \infty$,*

$$\sqrt{n}\left(U_{\mathrm{AIPW}}^{\hat{\pi}_\tau} - \theta^*\right) \xrightarrow{d} \mathcal{N}(0, \sigma_*^2),$$

*where $\sigma_*^2 = r^2 \operatorname{Var}\left(h_1^\mu(\hat{Y}) + [h_1(Y) - h_1^\mu(\hat{Y})]\frac{\xi_\tau^*}{\pi_\tau^*(X)}\right)$ with $\xi_\tau^* \sim \operatorname{Ber}(\pi_\tau^*(X))$. Consequently, for any $\hat{\sigma}^2 \xrightarrow{p} \sigma_*^2$, the confidence interval*

$$\mathcal{C}_\alpha = \left(\hat{\theta} \pm z_{1-\alpha/2}\,\hat{\sigma}/\sqrt{n}\right)$$

*is a valid $(1 - \alpha)$-confidence interval in the sense that $\lim_{n\to\infty}\mathbb{P}(\theta^* \in \mathcal{C}_\alpha) = 1 - \alpha$.*

The key technical ingredient is the common-random-number (CRN) coupling operation (Mohamed et al., 2020), which compares the relevant stochastic processes under shared randomness. This coupling strategy leads to a formalized proof route for active statistical inference.

**Variance estimation** For $i \in [n]$, let $\hat{h}_{1i} = h_1^\mu(\hat{Y}_i)$, $h_{1i} = h_1(Y_i)$, and let $\xi_i \in \{0, 1\}$ denote the sampling/labeling indicator with inclusion probability $\pi_i$. Our variance estimator mirrors the asymptotic variance decomposition in (5). Specifically, we estimate the CLT variance parameter $\sigma_*^2$ by

$$\hat{\sigma}^2 = \widehat{\operatorname{Var}}_1 + \widehat{\operatorname{Var}}_2. \tag{8}$$

where both components use Horvitz–Thompson (HT) corrections to account for missing labels. Equivalently, $\hat{\sigma}^2$ can be viewed as an estimator of $n\operatorname{Var}(U_{\mathrm{AIPW}}^\pi)$. We first estimate the mean and second moment of $h_{1i}$ using

$$\hat{m}_1 = \frac{1}{n}\sum_{i=1}^n \frac{\xi_i}{\pi_i}\,h_{1i}, \qquad \hat{m}_2 = \frac{1}{n}\sum_{i=1}^n \frac{\xi_i}{\pi_i}\,h_{1i}^2,$$

and define $\widehat{\operatorname{Var}}_{\mathrm{HT}}(h_1) = \hat{m}_2 - \hat{m}_1^2$. The first variance component is then estimated by

$$\widehat{\operatorname{Var}}_1 = r^2\,\widehat{\operatorname{Var}}_{\mathrm{HT}}(h_1).$$

The second term captures the additional variance induced by subsampling. We estimate $\sum_{i=1}^n (h_{1i} - \hat{h}_{1i})^2(1/\pi_i - 1)$ via its HT analogue, yielding

$$\widehat{\operatorname{Var}}_2 = \frac{r^2}{n}\sum_{i=1}^n \frac{\xi_i}{\pi_i}\,(h_{1i} - \hat{h}_{1i})^2\left(\frac{1}{\pi_i} - 1\right).$$

**Theorem 2.5** (Consistency of the variance estimator). *Under the conditions of Theorem 2.4, the variance estimator $\hat{\sigma}^2$ defined above is consistent for the CLT variance parameter, i.e., $\hat{\sigma}^2 \xrightarrow{p} \sigma_*^2$, as $n \to \infty$.*

## 3. Active Inference for $U$-estimation

In this section, we extend our active inference framework to $U$-statistics-based empirical risk minimization (ERM), or $U$-estimation, where the target parameter is defined as the minimizer of a population $U$-risk:

$$\theta^* = \arg\min_{\theta\in\Theta} L_0(\theta), \text{ where } L_0(\theta) = \mathbb{E}[\ell(\theta; Z_1, \ldots, Z_r)].$$

Here, the loss function $\ell(\theta; Z_1, \ldots, Z_r)$ is symmetric in $Z_1, \ldots, Z_r$ and $Z = (X, Y)$. Then, the $U$-estimator is defined as the minimizer of the empirical $U$-risk: $\hat{\theta} = \arg\min_{\theta\in\Theta} L_n(\theta)$, where

$$L_n(\theta) = \frac{1}{\binom{n}{r}}\sum_{\mathcal{C}_{n,r}} \ell(\theta; Z_{i_1}, \ldots, Z_{i_r}).$$

Through appropriate choices of the kernel function, the $U$-estimation framework unifies a wide range of problems in regression, classification, ranking, metric learning, and ROC analysis, allowing many modern machine learning objectives to be formulated as $U$-estimation problems.

*Example* 3.1 (Pairwise ranking). A canonical example is the ranking problem (Clémençon et al., 2008; Liu, 2009). The goal is to determine the ordering of $Y$ based upon the observed features $X$ by finding a rule $f : \mathcal{X} \times \mathcal{X} \to \mathbb{R}$ such that $Y_1$ is better than $Y_2$ if $f(X_1, X_2) > 0$. For linear ranking rules, $f(X_1, X_2) = \theta^\top(X_1 - X_2)$. We seek the optimal $\theta$ by introducing the following loss function:

$$\ell(\theta; Z_1, Z_2) = \phi(\operatorname{sign}(Y_1 - Y_2)\theta^\top(X_1 - X_2)),$$

where $\phi$ can be chosen as logistic loss $\phi(x) = \ln(1 + e^{-x})$.

Prior work in this area has largely focused on data-rich settings, developing scalable ERM and distributed optimization for $U$-risks (Clémençon et al., 2016; Lei et al., 2021; Chen et al., 2023). In contrast, we study the label-limited regime, where acquiring labels is costly. In this setting, directly computing $L_n(\theta)$ is infeasible, since most labels are unobserved. We therefore propose an active estimator for $\mathbb{E}[\ell(\theta; Z_1, \ldots, Z_r)]$ that combines machine learning predictions with adaptively queried true labels to enable efficient and statistically valid $U$-estimation.

Fix $\theta$, the loss $\ell(\theta; Z_1, \ldots, Z_r)$ then act as a $U$-statistic kernel. Let $\hat{Z} = (X, \hat{Y})$ denote pseudo-labeled data obtained from a predictive model. We define the active $U$-risk

$$L_{n,\pi}^{\mathrm{act}}(\theta) = \frac{1}{\binom{n}{r}}\sum_{\mathcal{C}_{n,r}} \ell(\theta; \hat{Z}_{i_1}, \ldots, \hat{Z}_{i_r})$$
$$+ \frac{1}{\binom{n}{r}}\sum_{\mathcal{C}_{n,r}} \Delta_l(\theta; i_1, \ldots, i_r)\frac{\xi_{i_1}\cdots\xi_{i_r}}{\pi_{i_1}\cdots\pi_{i_r}}, \tag{9}$$

with $\Delta_l(\theta; i_1, \ldots, i_r) = \ell(\theta; Z_{i_1}, \ldots, Z_{i_r}) - \ell(\theta; \hat{Z}_{i_1}, \ldots, \hat{Z}_{i_r})$. The first term uses predicted labels and is generally biased, while the second term provides

an IPW correction based on actively acquired labels. This yields an unbiased estimator of the population $U$-risk. The resulting active $U$-estimator is $\hat{\theta}_{act}^{\pi} = \arg\min_{\theta \in \Theta} L_{n,\pi}^{\mathrm{act}}(\theta)$.

### 3.1. Optimal sampling rule for active $U$-estimation

To understand how labels should be sampled, we analyze the asymptotic behavior of $U$-estimation. Under regularity conditions (Dang et al., 2008) (Assumption A.2 in Appendix A.1), the classical $U$-estimator $\hat{\theta}$ admits the expansion

$$\hat{\theta} - \theta^* = [\nabla^2 L_0(\theta^*)]^{-1} \times \frac{r}{n} \sum_{i=1}^{n} g(\theta^*; Z_i) + o_p(n^{-1/2}),$$
(10)

where $g(\theta; z) = \mathbb{E}[\nabla \ell(\theta; Z_1, \ldots, Z_r) | Z_1 = z]$ is the first-order projection of the gradient kernel and $\nabla^2 L_0(\theta^*)$ is the Hessian matrix. Since the first term dominates asymptotically, we can have the asymptotic normality

$$\sqrt{n}(\hat{\theta} - \theta^*) \to \mathcal{N}\left(0, r^2 [\nabla^2 L_0(\theta^*)]^{-1} \Sigma_g [\nabla^2 L_0(\theta^*)]^{-1}\right)$$

where $\Sigma_g = \mathrm{Cov}(g(\theta^*; Z))$.

Define $g^{\mu}(\theta; \hat{z}) = \mathbb{E}[\nabla \ell(\theta; \hat{Z}_1, \ldots, \hat{Z}_r) | \hat{Z}_1 = \hat{z}]$. Since $L_n^{\mathrm{act}}(\theta)$ in (9) is a linear combination of IPW correction terms, an analogous argument of (4) and (10) yields

$$\hat{\theta}_{act}^{\pi} - \theta^* = [\nabla^2 L_0(\theta^*)]^{-1} \times \frac{r}{n} \sum_{i=1}^{n} \phi(\theta^*; Z_i, \hat{Z}_i, \xi_i, \pi_i)$$
$$+ o_p(n^{-1/2}), \tag{11}$$

where $\phi(\theta^*; Z_i, \hat{Z}_i, \xi_i, \pi_i) = g(\theta^*; Z_i) + [g(\theta^*; Z_i) - g^{\mu}(\theta^*; \hat{Z}_i)] (\xi_i/\pi_i - 1)$. As a result,

$$\mathrm{Cov}(\hat{\theta}_{act}^{\pi}) \approx \frac{r^2}{n} [\nabla^2 L_0(\theta^*)]^{-1} \Sigma_g^{\pi} [\nabla^2 L_0(\theta^*)]^{-1},$$

where $\Sigma_g^{\pi}$ is the covariance matrix of $\phi(\theta^*; Z_i, \hat{Z}_i, \xi_i, \pi_i)$.

We seek a sampling policy $\pi$ that minimizes the variability of the estimator. To reduce the matrix objective to a scalar criterion, we adopt A-optimality, i.e., minimizing the trace of the covariance matrix $\mathrm{tr}(\mathrm{Cov}(\hat{\theta}_{act}^{\pi}))$ (Kiefer, 1959), a standard choice in optimal subsampling and prediction-powered inference (Wang et al., 2018; Angelopoulos et al., 2023b). By suitable calculation, we have

$$\mathrm{tr}(\mathrm{Cov}(\hat{\theta}_{act}^{\pi})) \propto \mathbb{E}\left[ \frac{1}{\pi(X)} \mathrm{tr}(A(\theta^*; X)[\nabla^2 L_0(\theta^*)]^{-2}) \right],$$

where $A(\theta^*; X) = \mathbb{E}\{[g(\theta^*; Z_i) - g^{\mu}(\theta^*; \hat{Z}_i)][g(\theta^*; Z_i) - g^{\mu}(\theta^*; \hat{Z}_i)]^{\top} | X\}$. A standard Lagrange multiplier argument suggests that to achieve smallest trace,

$$\pi(X) \propto \sqrt{\mathbb{E}[\mathrm{tr}(A(\theta^*; X)[\nabla^2 L_0(\theta^*)]^{-2}) | X]}.$$

**Theorem 3.2** (Optimal sampling probabilities for active $U$-estimation)**.** *Fix a labeling budget $n_b$. Let*

$$S(X) := \mathbb{E}\left[\mathrm{tr}(A(\theta^*; X)[\nabla^2 L_0(\theta^*)]^{-2}) \,|\, X\right],$$

*and set $\pi^*(X) = \min\{1, (n_b/n)(\sqrt{S(X)}/\mathbb{E}[\sqrt{S(X)}])\}$. Suppose Assumption A.2 holds. Among all sampling rules $\pi : \mathcal{X} \to [0, 1]$ with $\mathbb{E}\{\pi(X)\} \leq n_b/n$, we have that as $n \to \infty$*

$$\mathrm{tr}\left(\mathrm{Cov}\left(\hat{\theta}_{act}^{\pi}\right)\right) \geq \mathrm{tr}\left(\mathrm{Cov}\left(\hat{\theta}_{act}^{\pi^*}\right)\right).$$

Since the oracle sampling probability depends on the unknown $\theta^*$, we adopt a two-stage procedure:

(I) **Pilot estimator**: A small labeled subset is used to compute a pilot estimator $\theta^{\mathrm{pilot}}$, which serves as a plug-in surrogate for $\theta^*$ when constructing sampling probabilities.

(II) **Active ERM**: Based on a small amount of queried labels, we train a predictive model $S(\cdot)$ to predict $\mathrm{tr}\big((g(\theta^{\mathrm{pilot}}; Z_i) - g^{\mu}(\theta^{\mathrm{pilot}}; \hat{Z}_i))(g(\theta^{\mathrm{pilot}}; Z_i) - g^{\mu}(\theta^{\mathrm{pilot}}; \hat{Z}_i))^{\top}[\nabla^2 L_n(\theta^{\mathrm{pilot}})]^{-2}\big)$ based on $X_i$. Then the sampling probability $\hat{\pi}(X_i) = n_b(\sqrt{S(X_i)}/\sum_{j=1}^{n} \sqrt{S(X_j)})$. Through the sampling probability, we minimize the active loss function in (9) to obtain the final estimator.

Under regularity conditions, we can verify the asymptotic normality of the active $U$-estimator.

**Theorem 3.3** (CLT for Active $U$-estimation)**.** *Under Assumptions A.1 and A.2, we have $\sqrt{n}(\hat{\theta}_{\mathrm{act}}^{\hat{\pi}} - \theta^*) \xrightarrow{d} \mathcal{N}(0, \Sigma_*)$, where $\Sigma_* = r^2 [\nabla^2 L_0(\theta^*)]^{-1} \Sigma_g^{\pi} [\nabla^2 L_0(\theta^*)]^{-1}$. and $\Sigma_g^{\pi}$ is the covariance matrix of $\phi(\theta^*; Z_i, \hat{Z}_i, \xi_i^*, \pi_i)$ and $\xi^* \sim \mathrm{Ber}(\pi^*(X))$.*

Overall, this strategy highlights a key advantage of our approach: by exploiting the $U$-risks structure, labels are actively queried for efficient $U$-statistics-based ERM. We provide experiment evaluation in Appendix C.5.

## 4. Experiments

We evaluate the performance of our proposed active inference method using real-world datasets. On each dataset, we compare with the following intuitive benchmark methods:

- **classical**: The IPW $U$-statistic with uniform sampling, $U_{\mathrm{IPW}}^{\pi}(\mathcal{Y}_n)$ in (2) with $\pi(x) \equiv n_b/n$. To stabilize it, we replace the denominator $\binom{n}{r}$ with $\hat{N}$ in (6).

- **uniform**: The AIPW $U$-statistic $U_{\mathrm{AIPW}}^{\pi}(\mathcal{Y}_n)$ in (7) with uniform sampling, i.e., $\pi(x) \equiv n_b/n$.

- **active**: The proposed active $U$-statistic in Algorithm 1.

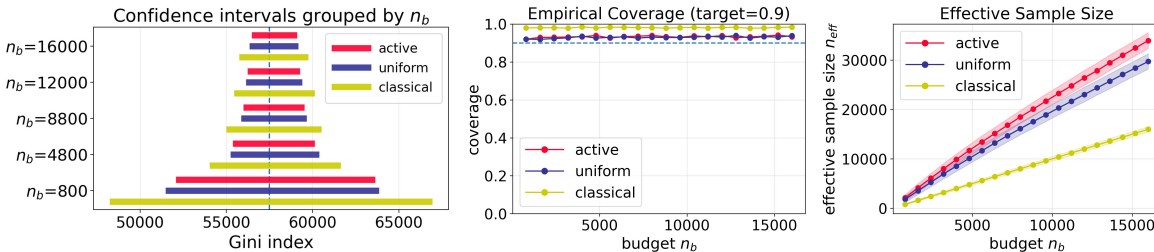

*Figure 1.* Income (ACS) dataset. Estimation of the Gini index as a measure of population income inequality. **Left**: 90% confidence intervals for the Gini index estimate at budgets $n_b \in \{800, 4800, 8800, 12000, 16000\}$. **Middle**: empirical coverage of the intervals (dashed line denotes the target coverage level 90%). **Right**: effective sample size $n_{\text{eff}}$, with shaded $\pm 1$ standard-deviation bands.

Each dataset uses a different base predictive model $f$, specified in the experiment. For each budget $n_b$, we report 90% confidence intervals for the target parameter, averaged over multiple replicates, to summarize estimation accuracy and uncertainty. We also report empirical coverage rates for the confidence intervals to assess the inferential validation.

We further compare methods using *effective sample size*. For an estimator run with labeling budget $n_b$, its effective sample size $n_{\text{eff}}$ is defined as the budget at which the classical estimator attains the same variance. Equivalently, an estimator has effective sample size $n_{\text{eff}}$ if its variance under budget $n_b$ matches the variance of the **classical** estimator with budget $n_{\text{eff}}$. A larger $n_{\text{eff}}$ indicates a more efficient estimator, while $n_{\text{eff}} < n_b$ implies that the estimator performs worse than the classical baseline. We report one standard deviation band around the effective sample size in all plots.

### 4.1. Income dataset (ACS)

We consider the **American Community Survey (ACS)** data (Ding et al., 2021) released by the U.S. Census Bureau, which consists of rich demographic and socioeconomic covariates, including age, education, marital status, employment, and occupation indicators. Each observation corresponds to an individual, and the label $Y$ represents the income measure. For this dataset, both the prediction model $\mu(X)$ and the proxy score model $V(x)$ are implemented using XGBoost: $\mu(X)$ is trained to predict the income label $Y$ from the demographic and socioeconomic covariates, while $V(x)$ is learned to construct the active sampling probabilities.

**Target parameter: Gini index**   Beyond point prediction of individual income, a central policy-relevant goal is to quantify income inequality in the population. The Gini index (Yitzhaki, 2003) is a standard measure of imbalance in the income distribution, defined as

$$\theta^* = \mathbb{E}[\,|Y_1 - Y_2|\,],$$

which corresponds to a $U$-statistic with kernel $h(y_1, y_2) = |y_1 - y_2|$. Estimating this parameter with uncertainty quan-

tification is crucial for downstream socioeconomic analysis. In practice, reliable income labels are expensive and difficult to obtain, often requiring additional survey or linkage to administrative records under strict access control. This motivates the need for active label acquisition.

Figure 1 compares the proposed active sampling method with the classical and uniform baselines. All methods achieve near-nominal coverage, while our active approach yields the narrowest confidence intervals. Across all budgets, active sampling attains a markedly larger effective sample size, corresponding to roughly a 60% labeling-budget reduction relative to classical one. It demonstrates the higher efficiency of our approach.

### 4.2. Perioperative dataset (VitalDB)

We further validate our approach on **VitalDB** (Lee et al., 2022), a perioperative dataset with records from 6,388 surgical cases, including case-level covariates (e.g., demographics and preoperative assessments) and time-stamped lab measurements (e.g., lactate, hemoglobin, blood gases).

We take the anesthesia start time $t_0$ as the index event and focus on hemoglobin. For each case, within a $\pm 24$ hour window around $t_0$, we define $Y^a$ as the most recent pre-event hemoglobin measurement ($t_0 - 24\text{h} < \texttt{dt} < t_0$) and $Y^b$ as the earliest post-event measurement ($t_0 < \texttt{dt} < t_0 + 24\text{h}$). The outcome of interest is the paired difference $D = Y^a - Y^b$. We will retain cases only for which both measurements are available. For this dataset, both the prediction model $\mu(X)$ and the proxy score model $V(x)$ are implemented using XGBoost: $\mu(X)$ is trained to predict the paired difference $D$ from case-level covariates, while $V(x)$ is learned to construct the active sampling probabilities.

**Target $U$-statistic: Wilcoxon signed-rank**   Accessing whether hemoglobin exhibits a *systematic shift* around anesthesia induction is important for monitoring perioperative management. The Wilcoxon signed-rank test (Wilcoxon, 1945) is well suited to this task: it provides a rank-based summary of directional change while reducing sensitivity to

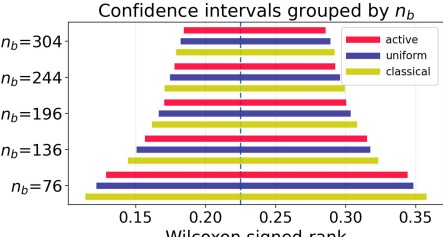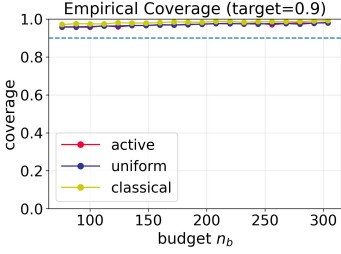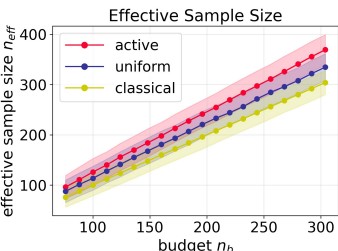

*Figure 2.* Perioperative dataset (VitalDB). Estimation of the Wilcoxon signed-rank test statistic to assess whether hemoglobin exhibits a systematic shift at anesthesia induction. **Left**: 90% confidence intervals for the estimate at budgets $n_b \in \{76, 136, 196, 244, 304\}$. **Middle**: empirical coverage of the intervals. **Right**: effective sample size $n_{\text{eff}}$, with shaded $\pm 1$ standard-deviation bands.

extreme or noisy measurements.

The Wilcoxon signed-rank test is asymptotically equivalent to the test based on the $U$-statistic

$$U_n^{(1)} = \binom{n}{2}^{-1} \sum_{1 \leq i < j \leq n} \mathbb{I}(D_i + D_j > 0).$$

This $U$-statistic is an unbiased estimator of the parameter

$$\theta^* = \mathbb{P}(D_1 + D_2 > 0).$$

Although VitalDB is covariate-rich, constructing $D$ requires two hemoglobin measurements tightly localized around $t_0$, so only a subset of cases yield valid pairs. This creates a label-limited setting that motivates active acquisition.

Figure 2 reports the confidence interval, empirical coverage and effective sample size, respectively. Across budgets, active sampling consistently achieves shorter intervals and larger $n_{\text{eff}}$ than classical and uniform baselines while maintaining near-nominal coverage. Specifically, it achieves comparable precision with about 20% fewer labels than classical baseline and 10% fewer than uniform sampling.

### 4.3. Political bias dataset

We further evaluate our method on a **political-bias** task using online news articles annotated for political leaning. Following the setup in Gligorić et al. (2025); Li et al. (2025), each sample consists of article text and metadata (topic, source, title, date, URL) with a ground-truth label

$$Y^{\text{TR}} \in \{\texttt{left}, \texttt{center}, \texttt{right}\}$$

We also obtain LLM-based proxy labels $Y^{\text{LLM}}$ and confidence scores, reflecting a label-limited setting where high-quality annotations are costly.

**Target parameter: Kendall's $\tau$**  Our goal is to assess whether LLM predictions preserve the relative ideological ordering of articles. To this end, we measure the agreement between the true label $Y^{\text{TR}}$ and the LLM prediction $Y^{\text{LLM}}$ from GPT-3.5 using Kendall's $\tau$ coefficient

(Kendall, 1948), a classical second-order $U$-statistic. After encoding the ordered categories as an ordinal score (e.g., $\texttt{left} < \texttt{center} < \texttt{right}$), the target parameter can be written as

$$\theta^* = \mathbb{E}\Big[\text{sign}\Big((Y_1^{\text{TR}} - Y_2^{\text{TR}})(Y_1^{\text{LLM}} - Y_2^{\text{LLM}})\Big)\Big].$$

Near-zero $\theta^*$ indicates frequent pairwise reversals, implying that LLM-based labels may distort ordinal structure and mislead downstream audits.

Figure 3 reports confidence intervals, empirical coverage, and effective sample size for the Kendall's $\tau$ task. The proposed active inference procedure remains well performed under this target. Across all labeling budgets, active sampling yields shorter intervals and larger $n_{\text{eff}}$ than the comparing benchmarks while maintaining near-nominal coverage. It requires approximately 20% less budget than classical baseline to achieve the same inferential accuracy, and about 10% less budget than uniform sampling. The estimated $\theta^*$ is moderately positive (about 0.3–0.4), suggesting the LLM captures coarse ideological ordering but still misorders a nontrivial fraction of article pairs.

## 5. Discussion

We point out several promising future directions. First, our current implementation leverages predictions from all unlabeled samples to construct augmented estimators, which may become computationally demanding in large-scale settings. An important extension is to design additional sampling or sketching strategies over the unlabeled data to reduce computational cost without sacrificing statistical efficiency, or to develop distributed algorithms for scalable estimation. Second, while our theoretical analysis focuses on asymptotic validity and efficiency, obtaining formal finite-sample guarantees for active $U$-statistics with learned sampling policies remains an important direction. Third, our analysis focuses on non-degenerate $U$-statistics, where the first-order projection governs the asymptotic behavior. When the first-order projection vanishes, higher-order components dominate, and while our framework can still be

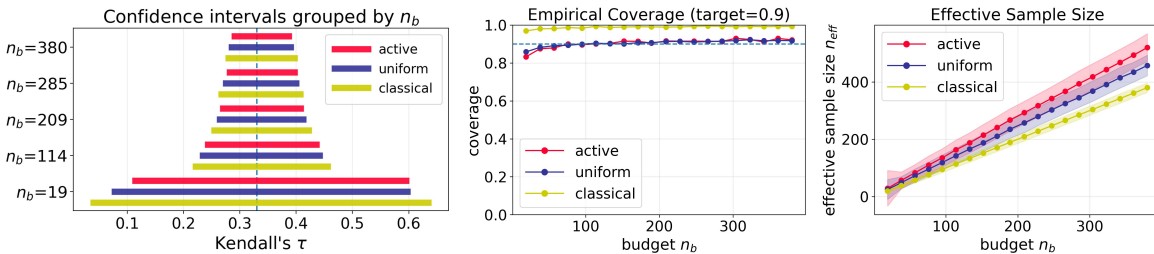

*Figure 3.* Political bias dataset. Estimation of Kendall's $\tau$ to detect whether LLM-based labels distort the underlying ordinal structure and thereby bias downstream audits. **Left**: 90% confidence intervals for estimate at budgets $n_b \in \{19, 114, 209, 285, 380\}$. **Middle**: empirical coverage of the intervals. **Right**: effective sample size $n_{\text{eff}}$, with shaded $\pm 1$ standard-deviation bands.

applied with suboptimal policies, identifying optimal sampling rules in this degenerate regime remains an open and challenging problem.

## Acknowledgements

We thank anonymous area chair and reviewers for their helpful comments. Changliang Zou is supported by the National Key R&D Program of China (Grant No. 2022YFA1003800) and the National Natural Science Foundation of China (Grant Nos. 12231011); Liuhua Peng is supported by ARC (Grant No. LP240100101); Yuyang Huo is supported by the National Natural Science Foundation of China (No. 124B2016).

## Impact Statement

This paper presents work whose goal is to advance the field of Machine Learning. There are many potential societal consequences of our work, none which we feel must be specifically highlighted here.

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

# A. Technical proofs

## A.1. Assumptions

The following assumption on the sampling probability is used for establishing the asymptotic normality of both the active $U$-statistic and the active $U$-estimator.

**Assumption A.1** (Regularity conditions for sampling probability).

(H1) (*Uniform positivity of sampling probability*) There exists a constant $\pi_{\min} > 0$ such that $\pi^*(X) \geq \pi_{\min}$ a.s., and $\mathbb{P}(\inf_{1 \leq i \leq n} \hat{\pi}(X_i) \geq \pi_{\min}/2) \to 1$.

(H2) (*Sampling probability learning consistency*) $\Delta_n := \sup_{1 \leq i \leq n} |\hat{\pi}(X_i) - \pi^*(X_i)| = O_p(n^{-1/2})$.

(H1) in Assumption A.1 ensures that no observation is sampled with vanishing probability. Such lower-bound conditions are standard in active inference and importance-sampling–based estimation, as they prevent excessive variance inflation (Chen et al., 2025a). In practice, (H1) can be enforced deterministically via the proposed trimming procedure, under which $\pi_{\min} = (1 - \tau)n_b/n$. (H2) requires that the estimated sampling probability $\hat{\pi}$ uniformly approximates the optimal policy $\pi^*$. This condition is satisfied whenever the underlying $V(X)$ used to construct $\pi^*$ is estimated with sufficient accuracy. Similar uniform consistency assumptions are commonly adopted in the active inference literature to facilitate asymptotic analysis of active sampling procedures (Zrnic & Candès, 2024a; Chen et al., 2025a).

The next series of assumptions is used for the decomposition of the $U$-estimator in (10) and verifying its asymptotic normality. These conditions are standard in the related literature (Dang et al., 2008; Chen et al., 2023).

**Assumption A.2** (Regularity conditions for $U$-estimation).

(J1) (*Non-degeneracy*) $\mathbb{E}\big[\ell(\theta; Z_1, \ldots, Z_r)\big]^2 < \infty$ and $\mathbb{E}\big[\|\nabla_\theta \ell(\theta^*; Z_1, \ldots, Z_r)\|^2\big] < \infty$ for all $\theta \in \Theta$.

(J2) (*Continuity and compactness*) The map $\theta \mapsto \ell(\theta; z_1, \ldots, z_r)$ is continuous on $\Theta$ and $\Theta$ is compact.

(J3) (*Positive definite covariance matrix*) $H_0 := \nabla^2 L_0(\theta^*)$ is nonsingular and $\Sigma_g := \mathrm{Cov}\big(g(\theta^*; Z_1)\big)$ is positive definite.

(J4) (*Identification*) The population risk $L_0(\theta)$ admits a unique minimizer $\theta^* \in \Theta$.

(J5) (*Local Lipschitz of the gradient*) There exist $\delta > 0$ and a nonnegative random variable $L = L(Z_1, \ldots, Z_r)$ such that $\mathbb{E}[L^2] < \infty$ and, for all $\theta, \theta' \in \Theta$ with $\|\theta - \theta^*\| \leq \delta$ and $\|\theta' - \theta^*\| \leq \delta$,

$$\big\|\nabla_\theta \ell(\theta; Z_1, \ldots, Z_r) - \nabla_\theta \ell(\theta'; Z_1, \ldots, Z_r)\big\| \leq L \|\theta - \theta'\| \quad \text{a.s..}$$

## A.2. Useful Lemmas

**Lemma A.3.** *Let $\pi^*(x) \in (0, 1)$ be the target sampling probability, and let $\hat{\pi}(x)$ be a plug-in estimator. Assume a common-random-number (CRN) coupling: $U_i \overset{iid}{\sim} \mathrm{Unif}(0, 1)$ independent of $(X_i, Y_i, \hat{Y}_i)$ and*

$$\xi_i(\pi) := \mathbf{1}\{U_i \leq \pi(X_i)\}, \qquad \xi_i^* := \xi_i(\pi^*), \quad \hat{\xi}_i := \xi_i(\hat{\pi}).$$

*Denote $\omega_i$ be a measurable function w.r.t. $(X_i, Y_i, \hat{Y}_i)$ and satisfies $\mathbb{E}[\omega_1^2] < \infty$. Define the HT-type functional*

$$T_n(\pi) := \frac{1}{n} \sum_{i=1}^{n} \omega_i \frac{\xi_i(\pi)}{\pi(X_i)}.$$

*Suppose Assumption A.1 holds, then*

$$\sqrt{n}\big(T_n(\hat{\pi}) - T_n(\pi^*)\big) \overset{p}{\to} 0, \quad \text{and in fact} \quad T_n(\hat{\pi}) - T_n(\pi^*) = O_p(n^{-3/4}).$$

*Proof.* Write $\pi_i^* = \pi^*(X_i)$ and $\hat{\pi}_i = \hat{\pi}(X_i)$. Define

$$D_i := \frac{\hat{\xi}_i}{\hat{\pi}_i} - \frac{\xi_i^*}{\pi_i^*}, \qquad \text{so that} \qquad T_n(\hat{\pi}) - T_n(\pi^*) = \frac{1}{n} \sum_{i=1}^{n} \omega_i D_i.$$

Work on the event $\mathcal{E}_n = \{\inf_i \hat{\pi}_i \geq \pi_{\min}/2\}$, with $\mathbb{P}(\mathcal{E}_n) \to 1$ by (H1).

For any $\pi \in (0,1)$, $\mathbb{E}[\xi_i(\pi) \mid X_i] = \pi(X_i)$, hence $\mathbb{E}[\xi_i(\pi)/\pi(X_i) \mid X_i] = 1$. Therefore, $\mathbb{E}[D_i \mid X_i, \hat{\pi}] = 0$. Let $p = \pi_i^*$ and $q = \hat{\pi}_i$. Under the CRN construction $\xi = \mathbf{1}\{U \leq \cdot\}$, a direct interval-splitting calculation gives

$$\mathbb{E}[D_i^2 \mid X_i, \hat{\pi}] = \frac{|q - p|}{pq}.$$

On $\mathcal{E}_n$, $p \geq \pi_{\min}$ and $q \geq \pi_{\min}/2$, so

$$\mathbb{E}[D_i^2 \mid X_i, \hat{\pi}] \leq \frac{2}{\pi_{\min}^2} |\hat{\pi}_i - \pi_i^*| \leq \frac{2}{\pi_{\min}^2} \Delta_n.$$

Conditional on $(X_{1:n}, \hat{\pi}, Z_{1:n})$, the variables $\{D_i\}$ have conditional mean zero and are independent, since they are functions of independent $\{U_i\}$. Therefore,

$$\mathrm{Var}\left(\frac{1}{n} \sum_{i=1}^n \omega_i D_i \mid X_{1:n}, \hat{\pi}, \omega_{1:n}\right) = \frac{1}{n^2} \sum_{i=1}^n \omega_i^2 \, \mathbb{E}[D_i^2 \mid \omega_i, \hat{\pi}] \leq C \Delta_n \cdot \frac{1}{n^2} \sum_{i=1}^n \omega_i^2,$$

for $C = 2/\pi_{\min}^2$ on $\mathcal{E}_n$. Since $\mathbb{E}[\omega_1^2] < \infty$, we have $\frac{1}{n} \sum \omega_i^2 = O_p(1)$, hence

$$\mathrm{Var}\left(\frac{1}{n} \sum_{i=1}^n \omega_i D_i \mid X_{1:n}, \hat{\pi}, \omega_{1:n}\right) = O_p\left(\frac{\Delta_n}{n}\right).$$

By Chebyshev's inequality, this implies

$$\frac{1}{n} \sum_{i=1}^n \omega_i D_i = O_p\left(\sqrt{\frac{\Delta_n}{n}}\right).$$

Using $\Delta_n = O_p(n^{-1/2})$ from (H2), we obtain

$$T_n(\hat{\pi}) - T_n(\pi^*) = O_p(n^{-3/4}), \qquad \sqrt{n}\big(T_n(\hat{\pi}) - T_n(\pi^*)\big) = O_p(n^{-1/4}) \xrightarrow{P} 0.$$

$\square$

**Lemma A.4.** *Let $\{(X_i, W_i, \xi_i)\}_{i=1}^n$ be i.i.d., where $W_i$ may collect $(Y_i, \hat{Y}_i)$. Let $\pi^*(x) \in (0,1)$ be the target sampling probability and assume*

$$\xi_i \mid X_i \sim \mathrm{Ber}\big(\pi^*(X_i)\big), \qquad \{\xi_i\}_{i=1}^n \text{ are conditionally independent given } \{X_i\}_{i=1}^n.$$

*Let $\hat{\pi}(x)$ be a plug-in estimator and denote $\pi_i^* = \pi^*(X_i)$ and $\hat{\pi}_i = \hat{\pi}(X_i)$. Define*

$$T_n(\pi) := \frac{1}{n} \sum_{i=1}^n \xi_i \, \omega_i \, \eta\big(\pi(X_i)\big).$$

*Suppose Assumption A.1 holds, $\mathbb{E}[\omega_1^2] < \infty$ and $\eta$ is Lipschitz on $[\pi_{\min}/2, 1)$: there exists $L_\eta < \infty$ such that*

$$|\eta(u) - \eta(v)| \leq L_\eta |u - v|, \qquad \forall \, u, v \in [\pi_{\min}/2, 1).$$

*Then $T_n(\hat{\pi}) - T_n(\pi^*) = o_p(1)$.*

*Proof.* Work on the event $\mathcal{E}_n = \{\inf_i \hat{\pi}_i \geq \pi_{\min}/2\}$ with $\mathbb{P}(\mathcal{E}_n) \to 1$ by (H1). On $\mathcal{E}_n$, by (H2),

$$|T_n(\hat{\pi}) - T_n(\pi^*)| \leq \frac{1}{n} \sum_{i=1}^n \xi_i |\omega_i| \, \big|\eta(\hat{\pi}_i) - \eta(\pi_i^*)\big| \leq L_\eta \Delta_n \cdot \frac{1}{n} \sum_{i=1}^n \xi_i |Z_i|.$$

By $\mathbb{E}[\omega_1^2] < \infty$ and $0 \leq \xi_i \leq 1$, we have $\mathbb{E}[\xi_1 |\omega_1|] \leq \mathbb{E}[|\omega_1|] < \infty$, hence $\frac{1}{n} \sum_{i=1}^n \xi_i |\omega_i| = O_p(1)$ by the LLN. Since $\Delta_n = o_p(1)$ by (H2), it follows that $|T_n(\hat{\pi}) - T_n(\pi^*)| = o_p(1)$ on $\mathcal{E}_n$, hence unconditionally. $\square$

### A.3. Proof of Proposition 2.1

*Proof.* As $\xi_i \sim \mathrm{Bern}(\pi(X_i))$, $\mathbb{E}[\xi_i \mid X_i] = \pi(X_i) = \pi_i$ for $i \in [n]$. In addition, since $(\xi_{i_1}, \ldots, \xi_{i_r})$ are independent given $(X_{i_1}, \ldots, X_{i_r})$ for any $(i_1, \ldots, i_r) \in \mathcal{C}_{n,r}$, we have

$$\mathbb{E}\left[\frac{\xi_{i_1} \cdots \xi_{i_r}}{\pi_{i_1} \cdots \pi_{i_r}} \mid X_{i_1}, \ldots, X_{i_r}\right] = \frac{\mathbb{E}[\xi_{i_1} \mid X_{i_1}] \cdots \mathbb{E}[\xi_{i_r} \mid X_{i_r}]}{\pi_{i_1} \cdots \pi_{i_r}} = 1.$$

Therefore, conditioning on $(X_1, \ldots, X_n; Y_1, \ldots, Y_n)$,

$$\mathbb{E}[U_{\mathrm{AIPW}}^\pi \mid X_1, \ldots, X_n; Y_1, \ldots, Y_n]$$

$$= \frac{1}{\binom{n}{r}} \sum_{\mathcal{C}_{n,r}} \mathbb{E}\left[h(\hat{Y}_{i_1}, \ldots, \hat{Y}_{i_r}) \mid X_1, \ldots, X_n; Y_1, \ldots, Y_n\right] + \frac{1}{\binom{n}{r}} \sum_{\mathcal{C}_{n,r}} \mathbb{E}\left[\Delta_h(i_1, \ldots, i_r) \frac{\xi_{i_1} \cdots \xi_{i_r}}{\pi_{i_1} \cdots \pi_{i_r}} \mid X_1, \ldots, X_n; Y_1, \ldots, Y_n\right]$$

$$= \frac{1}{\binom{n}{r}} \sum_{\mathcal{C}_{n,r}} h(\hat{Y}_{i_1}, \ldots, \hat{Y}_{i_r}) + \frac{1}{\binom{n}{r}} \sum_{\mathcal{C}_{n,r}} \Delta_h(i_1, \ldots, i_r) \mathbb{E}\left[\frac{\xi_{i_1} \cdots \xi_{i_r}}{\pi_{i_1} \cdots \pi_{i_r}} \mid X_{i_1}, \ldots, X_{i_r}\right]$$

$$= \frac{1}{\binom{n}{r}} \sum_{\mathcal{C}_{n,r}} h(\hat{Y}_{i_1}, \ldots, \hat{Y}_{i_r}) + \frac{1}{\binom{n}{r}} \sum_{\mathcal{C}_{n,r}} \Delta_h(i_1, \ldots, i_r)$$

$$= \frac{1}{\binom{n}{r}} \sum_{\mathcal{C}_{n,r}} h(Y_{i_1}, \ldots, Y_{i_r}).$$

Taking expectation over $(X_1, \ldots, X_n; Y_1, \ldots, Y_n)$, we obtain

$$\mathbb{E}[U_{\mathrm{AIPW}}^\pi] = \mathbb{E}[h(Y_1, \ldots, Y_r)] = \theta^*.$$

$\square$

### A.4. Proof of Theorem 2.2

*Proof.* Under the conditions that $\{(X_i, Y_i)\}_{i=1}^n$ are i.i.d., $E[h(Y_1, \ldots, Y_r)]^2 < \infty$, $E[h(\hat{Y}_1, \ldots, \hat{Y}_r)]^2 < \infty$, $\mathrm{Var}(h_1(Y)) > 0$ and $\mathrm{Var}(h_1^\mu(\hat{Y})) > 0$, the decomposition in (4) gives

$$\mathrm{Var}(U_{\mathrm{AIPW}}^\pi) = C_n + \frac{r^2}{n} \mathbb{E}\left[s(X)\left(\frac{1}{\pi(X)} - 1\right)\right] + o(n^{-1}),$$

where $C_n$ does not depend on $\pi$. Thus, asymptotically minimizing $\mathrm{Var}(U_{\mathrm{AIPW}}^\pi)$ over $\pi$ satisfying $\mathbb{E}[\pi(X)] \le n_b/n$ is equivalent to minimizing

$$\mathbb{E}\left[\frac{s(X)}{\pi(X)}\right] \quad \text{subject to} \quad 0 \le \pi(X) \le 1, \ \mathbb{E}[\pi(X)] \le \frac{n_b}{n},$$

since $\mathbb{E}[s(X)]$ is constant in $\pi$.

Because $s(X)/\pi(X)$ is pointwise decreasing in $\pi(X)$, any minimizer must exhaust the labeling budget, and we may therefore restrict attention to policies satisfying $\mathbb{E}[\pi(X)] = n_b/n$. By the Cauchy–Schwarz inequality,

$$\mathbb{E}\left[\frac{s(X)}{\pi(X)}\right] = \mathbb{E}\left[\left(\frac{\sqrt{s(X)}}{\sqrt{\pi(X)}}\right)^2\right] \ge \frac{\left(\mathbb{E}[\sqrt{s(X)}]\right)^2}{\mathbb{E}[\pi(X)]} = \frac{\left(\mathbb{E}[\sqrt{s(X)}]\right)^2}{n_b/n}.$$

Equality holds if and only if $\sqrt{s(X)}/\sqrt{\pi(X)}$ is proportional to $\sqrt{\pi(X)}$ almost surely, which implies $\pi(X) \propto \sqrt{s(X)}$. Imposing the constraint $\pi(X) \le 1$ yields the optimal sampling rule

$$\pi^*(X) = \min\left\{1, \ c\sqrt{s(X)}\right\},$$

for some constant $c > 0$. Taking $c = (n_b/n)/\mathbb{E}[\sqrt{s(X)}]$ gives the stated form

$$\pi^*(X) = \min\left\{1, \ \frac{n_b}{n} \cdot \frac{\sqrt{s(X)}}{\mathbb{E}[\sqrt{s(X)}]}\right\},$$

which ensures $\mathbb{E}[\pi^*(X)] \leq n_b/n$.

Therefore, for any admissible sampling policy $\pi$,

$$\mathbb{E}[s(X)/\pi(X)] \geq \mathbb{E}[s(X)/\pi^*(X)].$$

Substituting this inequality into the variance expansion above yields

$$\mathrm{Var}(U_{\mathrm{AIPW}}^{\pi}) \geq \mathrm{Var}\left(U_{\mathrm{AIPW}}^{\pi^*}\right) \qquad \text{as } n \to \infty,$$

which completes the proof. $\qquad\square$

### A.5. Proof of Theorem 2.3

*Proof.* Denote

$$\hat{S} = \sum_{\mathcal{C}_{n,r}} \Delta_h(i_1, \ldots, i_r) \frac{\xi_{i_1} \cdots \xi_{i_r}}{\pi_{i_1} \cdots \pi_{i_r}},$$

then

$$U_{\mathrm{AIPW}}^{\pi,N} = \frac{1}{\binom{n}{r}} \sum_{\mathcal{C}_{n,r}} h\left(\hat{Y}_{i_1}, \ldots, \hat{Y}_{i_r}\right) + \frac{\hat{S}}{\widehat{N}}. \tag{12}$$

For any fixed $(i_1, \ldots, i_r) \in \mathcal{C}_{n,r}$,

$$\mathbb{E}\left[\frac{\xi_{i_1} \cdots \xi_{i_r}}{\pi_{i_1} \cdots \pi_{i_r}} \;\middle|\; X_{i_1}, \ldots, X_{i_r}\right] = 1.$$

Hence, $\mathbb{E}[\widehat{N}] = \binom{n}{r}$ and $\mathbb{E}[\hat{S}] = \binom{n}{r} \mathbb{E}[\Delta_h(1, \ldots, r)]$. Moreover, as $\mathbb{E}[\Delta_h(1, \ldots, r)]^2 < \infty$, we have $\widehat{N}/\binom{n}{r} \xrightarrow{p} 1$ and $\hat{S}/\binom{n}{r} \xrightarrow{p} \mathbb{E}[\Delta_h(1, \ldots, r)]$ by the law of large numbers. It follows that the normalized quantity satisfies:

$$\frac{\hat{S}}{\widehat{N}} = \frac{\hat{S}/\binom{n}{r}}{\widehat{N}/\binom{n}{r}} \xrightarrow{p} \mathbb{E}[\Delta_h(1, \ldots, r)].$$

It remains to show $\mathbb{E}[\hat{S}/\widehat{N}] = \mathbb{E}[\Delta_h(1, \ldots, r)] + o(1)$.

On the event $\{\widehat{N} \geq \frac{1}{2}\binom{n}{r}\}$, we have $|\hat{S}/\widehat{N}| \leq 2|\hat{S}|/\binom{n}{r}$. Under $\mathbb{E}[\Delta_h(1, \ldots, r)]^2 < \infty$ and $\pi_{\min} > 0$, the second moment of $\hat{S}/\binom{n}{r}$ is uniformly bounded in $n$. Hence, the sequence $\{\hat{S}/\binom{n}{r}\}_n$ is uniformly integrable. Moreover, $\widehat{N}/\binom{n}{r} \xrightarrow{p} 1$ implies $\mathbb{P}(\widehat{N} \geq \frac{1}{2}\binom{n}{r}) \to 1$. Therefore, $\{\hat{S}/\widehat{N}\}_n$ is uniformly integrable. Combining this with $\hat{S}/\widehat{N} \xrightarrow{p} \mathbb{E}[\Delta_h(1, \ldots, r)]$, we obtain

$$\mathbb{E}\left[\frac{\hat{S}}{\widehat{N}}\right] = \mathbb{E}[\Delta_h(1, \ldots, r)] + o(1).$$

Taking expectations on both sides of (12), we obtain

$$\mathbb{E}[U_{\mathrm{AIPW}}^{\pi,N}] = \theta^\mu + \mathbb{E}[\Delta_h(1, \ldots, r)] + o(1) = \theta^* + o(1),$$

which completes the proof. $\qquad\square$

### A.6. Proof of Theorem 2.4

*Proof.* For simplicity, we set $\tau = 1$. Then $\pi_\tau^* = \pi^*$. Let $\hat{\pi}(x)$ be the plug-in sampling probability, and adopt the CRN coupling in Lemma A.3: $\hat{\xi}_i := \xi_i(\hat{\pi})$ and $\xi_i^* := \xi_i(\pi^*)$. Plugging $\hat{\pi}$ and $\pi^*$ into (4), we obtain

$$U_{\mathrm{AIPW}}^{\hat{\pi}} - U_{\mathrm{AIPW}}^{\pi^*} = \frac{r}{n} \sum_{i=1}^{n} \left( h_1^\mu(\hat{Y}_i) + \{h_1(Y_i) - h_1^\mu(\hat{Y}_i)\} \frac{\hat{\xi}_i}{\hat{\pi}(X_i)} - \left[ h_1^\mu(\hat{Y}_i) + \{h_1(Y_i) - h_1^\mu(\hat{Y}_i)\} \frac{\xi_i^*}{\pi^*(X_i)} \right] \right) + o_p(n^{-1/2}). \tag{13}$$

Define
$$\psi_i(\pi) := h_1^\mu(\hat{Y}_i) + \{h_1(Y_i) - h_1^\mu(\hat{Y}_i)\}\frac{\xi_i(\pi)}{\pi(X_i)}.$$

The only dependence on $\pi$ is through the HT factor, and therefore

$$\frac{1}{n}\sum_{i=1}^n (\psi_i(\hat{\pi}) - \psi_i(\pi^*)) = \frac{1}{n}\sum_{i=1}^n \omega_i\left(\frac{\hat{\xi}_i}{\hat{\pi}(X_i)} - \frac{\xi_i^*}{\pi^*(X_i)}\right) = T_n(\hat{\pi}) - T_n(\pi^*), \tag{14}$$

where

$$\omega_i := h_1(Y_i) - h_1^\mu(\hat{Y}_i), \qquad T_n(\pi) := \frac{1}{n}\sum_{i=1}^n \omega_i\frac{\xi_i(\pi)}{\pi(X_i)}.$$

Then Lemma A.3 applies to $T_n(\cdot)$ and gives

$$\sqrt{n}\{T_n(\hat{\pi}) - T_n(\pi^*)\} \xrightarrow{p} 0.$$

Combining this with (13) and (14), we conclude that

$$\sqrt{n}\left(U_{\text{AIPW}}^{\hat{\pi}} - U_{\text{AIPW}}^{\pi^*}\right) \xrightarrow{p} 0. \tag{15}$$

Moreover, by construction $\{\psi_i(\pi^*)\}_{i=1}^n$ are i.i.d. with finite variance $\text{Var}(\psi^*)$, hence the classical CLT implies

$$\sqrt{n}\left(\frac{1}{n}\sum_{i=1}^n (\psi_i(\pi^*) - \mathbb{E}[\psi^*])\right) \xrightarrow{d} \mathcal{N}(0, \text{Var}(\psi^*)).$$

Since $\mathbb{E}[\psi^*] = \theta^*$, Slutsky's theorem together with (15) yields

$$\sqrt{n}(U_{\text{AIPW}}^{\hat{\pi}} - \theta^*) \xrightarrow{d} \mathcal{N}(0, r^2\text{Var}(\psi^*)),$$

where $\psi^* = h_1^\mu(\hat{Y}) + \{h_1(Y) - h_1^\mu(\hat{Y})\}\xi^*/\pi^*(X)$ and $\xi^* \mid X \sim \text{Ber}(\pi^*(X))$. For $\tau < 1$, the proof can be directly generalized as long as $\sup_{1 \le i \le n}|\hat{\pi}_\tau(X_i) - \pi_\tau^*(X_i)| = O_p(n^{-1/2})$. Notice that

$$|\hat{\pi}_\tau(X_i) - \pi_\tau^*(X_i)| = \tau|\hat{\pi}(X_i) - \pi^*(X_i)|.$$

By (H1) in Assumption A.1, the condition directly follows. This completes the proof.

$\square$

## A.7. Proof of Theorem 2.5

*Proof.* Let $\mu_1 = \mathbb{E}[h_1(Y)]$ and $\mu_2 = \mathbb{E}[h_1(Y)^2]$. We have

$$\hat{m}_1(\pi^*) \xrightarrow{p} \mu_1, \qquad \hat{m}_2(\pi^*) \xrightarrow{p} \mu_2, \qquad \hat{S}(\pi^*) \xrightarrow{p} \mathbb{E}\left[(h_1(Y) - h_1^\mu(\hat{Y}))^2\left(\frac{1}{\pi^*(X)} - 1\right)\right].$$

Thus
$$\widehat{\text{Var}}_{\text{HT}}^*(h_1) := \hat{m}_2(\pi^*) - \hat{m}_1(\pi^*)^2 \xrightarrow{p} \mu_2 - \mu_1^2 = \text{Var}(h_1(Y)).$$

By Lemma A.4, applied with $\eta(t) = t^{-1}$ and $Z_i = h_{1i}$ (resp. $Z_i = h_{1i}^2$), we obtain $\hat{m}_k(\hat{\pi}) - \hat{m}_k(\pi^*) = o_p(1)$ for $k = 1, 2$. Applying the same lemma with $\eta(t) = t^{-2} - t^{-1}$ and $Z_i = (h_{1i} - \hat{h}_{1i})^2$ yields $\hat{S}(\hat{\pi}) - \hat{S}(\pi^*) = o_p(1)$. Hence $\hat{m}_k(\hat{\pi}) \xrightarrow{p} \mu_k$ and $\hat{S}(\hat{\pi}) \xrightarrow{p} S^*$, and

$$\widehat{\text{Var}}_{\text{HT}}(h_1) = \hat{m}_2(\hat{\pi}) - \hat{m}_1(\hat{\pi})^2 \xrightarrow{p} \text{Var}(h_1(Y)).$$

Therefore,
$$\widehat{\text{Var}}_1 = r^2\widehat{\text{Var}}_{\text{HT}}(h_1) \xrightarrow{p} r^2\text{Var}(h_1(Y)), \qquad \widehat{\text{Var}}_2 = r^2\hat{S}(\hat{\pi}) \xrightarrow{p} r^2 S^*.$$

By the standard AIPW variance decomposition (using $\mathbb{E}[\xi^*/\pi^*(X) \mid X] = 1$),

$$\text{Var}(\psi^*) = \text{Var}(h_1(Y)) + \mathbb{E}\left[(h_1(Y) - h_1^\mu(\hat{Y}))^2\left(\frac{1}{\pi^*(X)} - 1\right)\right].$$

Thus $\sigma_*^2 = r^2\text{Var}(\psi^*)$ equals the sum of the two limits above, and hence $\widehat{\text{Var}}_1 + \widehat{\text{Var}}_2 \xrightarrow{p} \sigma_*^2$. $\square$

## A.8. Proof of Theorem 3.2

*Proof.* Suppose Assumption A.2 holds, it has

$$\mathrm{tr}\Big(\mathrm{Cov}(\hat{\theta}^\pi_{\mathrm{act}})\Big) = C_n + \frac{r^2}{n}\,\mathbb{E}\bigg[\frac{1}{\pi(X)}\,\mathrm{tr}\big(A(\theta^*;X)[\nabla^2 L_0(\theta^*)]^{-2}\big)\bigg] + o(n^{-1}),$$

where $C_n$ does not depend on $\pi$. Let

$$S(X) := \mathbb{E}\big[\mathrm{tr}\big(A(\theta^*;X)[\nabla^2 L_0(\theta^*)]^{-2}\big)\,\big|\,X\big] \geq 0.$$

Hence, minimizing $\mathrm{tr}(\mathrm{Cov}(\hat{\theta}^\pi_{\mathrm{act}}))$ over $\pi : \mathcal{X} \to [0,1]$ with $\mathbb{E}\{\pi(X)\} \leq n_b/n$ is asymptotically equivalent to minimizing

$$\mathbb{E}\bigg[\frac{S(X)}{\pi(X)}\bigg] \quad \text{s.t.} \quad 0 \leq \pi(X) \leq 1, \ \ \mathbb{E}\{\pi(X)\} \leq \frac{n_b}{n}.$$

Since $s(X)/\pi(X)$ is pointwise decreasing in $\pi(X)$, any minimizer uses the full budget, so we impose $\mathbb{E}\{\pi(X)\} = n_b/n$.

Consider the Lagrangian

$$\mathcal{L}(\pi,\lambda) = \mathbb{E}\bigg[\frac{S(X)}{\pi(X)} + \lambda\,\pi(X)\bigg] - \lambda\frac{n_b}{n}, \qquad \lambda \geq 0.$$

For fixed $\lambda$, the integrand is separable in $X$, so the minimizer is obtained pointwise by minimizing $u \mapsto S(X)/u + \lambda u$ over $u \in (0,1]$. The unconstrained minimizer satisfies $-S(X)/u^2 + \lambda = 0$, hence $u = \sqrt{S(X)/\lambda}$. Imposing $u \leq 1$ yields

$$\pi_\lambda(X) = \min\Big\{1, \sqrt{S(X)/\lambda}\Big\}.$$

Choose $\lambda = \lambda^*$ so that $\mathbb{E}\{\pi_{\lambda^*}(X)\} = n_b/n$. Writing $c = 1/\sqrt{\lambda^*}$ gives

$$\pi^*(X) = \min\{1, c\sqrt{S(X)}\} = \min\bigg\{1, \frac{n_b}{n}\cdot\frac{\sqrt{S(X)}}{\mathbb{E}[\sqrt{S(X)}]}\bigg\}.$$

By KKT optimality, $\pi^*$ minimizes $\mathbb{E}[S(X)/\pi(X)]$ over all admissible $\pi$, and plugging this back into the expansion above gives

$$\mathrm{tr}\Big(\mathrm{Cov}\big(\hat{\theta}^\pi_{act}\big)\Big) \geq \mathrm{tr}\Big(\mathrm{Cov}\big(\hat{\theta}^{\pi^*}_{act}\big)\Big) \qquad \text{as } n \to \infty.$$

$\square$

## A.9. Proof of Theorem 3.3

*Proof.* Since (11) holds with any sampling rule $\pi(\cdot)$ and that $\{\phi(\theta^*;Z_i,\hat{Z}_i,\xi_i,\pi_i)\}_{i=1}^n$ are i.i.d. with

$$\mathbb{E}\Big[\phi(\theta^*;Z,\hat{Z},\xi,\pi)\Big] = 0, \qquad \mathbb{E}\Big[\|\phi(\theta^*;Z,\hat{Z},\xi,\pi)\|^2\Big] < \infty,$$

and define $\Sigma^\pi_g := \mathrm{Var}\Big(\phi(\theta^*;Z,\hat{Z},\xi,\pi)\Big)$. Under the regularity conditions in Assumption A.2,

$$\sqrt{n}\,(\hat{\theta}^\pi_{\mathrm{act}} - \theta^*) \xrightarrow{d} \mathcal{N}\bigg(0,\ r^2\Big[\nabla^2 L_0(\theta^*)\Big]^{-1}\Sigma^\pi_g\Big[\nabla^2 L_0(\theta^*)\Big]^{-1}\bigg). \tag{16}$$

Since the oracle linear representation (11) holds for $\pi = \pi^*$ and $\pi = \hat{\pi}$:

$$\hat{\theta}^\pi_{\mathrm{act}} - \theta^* = \frac{r}{n}\sum_{i=1}^n \phi(\theta^*;Z_i,\hat{Z}_i,\xi_i(\pi),\pi_i)\,H_0^{-1} + o_p(n^{-1/2}), \qquad H_0 := \nabla^2 L_0(\theta^*).$$

Subtracting the two expansions gives

$$\hat{\theta}^{\hat{\pi}}_{\mathrm{act}} - \hat{\theta}^{\pi^*}_{\mathrm{act}} = \frac{r}{n}\sum_{i=1}^n \Big\{\phi(\theta^*;Z_i,\hat{Z}_i,\hat{\xi}_i,\hat{\pi}_i) - \phi(\theta^*;Z_i,\hat{Z}_i,\xi^*_i,\pi^*_i)\Big\}H_0^{-1} + o_p(n^{-1/2}). \tag{17}$$

By the definition of $\phi$, the $g(\theta^*; Z_i)$ term cancels and we can write

$$\phi(\theta^*; Z_i, \hat{Z}_i, \hat{\xi}_i, \hat{\pi}_i) - \phi(\theta^*; Z_i, \hat{Z}_i, \xi_i^*, \pi_i^*) = \omega_i \left( \frac{\hat{\xi}_i}{\hat{\pi}_i} - \frac{\xi_i^*}{\pi_i^*} \right), \qquad \omega_i := g(\theta^*; Z_i) - g^\mu(\theta^*; \hat{Z}_i).$$

By the moment assumption $\mathbb{E}\|g(\theta^*; Z)\|^2 + \mathbb{E}\|g^\mu(\theta^*; \hat{Z})\|^2 < \infty$, we have $\mathbb{E}\|\omega_1\|^2 < \infty$, the requirement $\mathbb{E}[\omega_1^2] < \infty$ in Lemma A.3 is satisfied. By Lemma A.3, we have

$$\hat{\theta}_{\mathrm{act}}^{\hat{\pi}} - \hat{\theta}_{\mathrm{act}}^{\pi^*} \xrightarrow{p} 0.$$

Plugging this into (16) yields

$$\sqrt{n}\big(\hat{\theta}_{\mathrm{act}}^{\hat{\pi}} - \hat{\theta}_{\mathrm{act}}^{\pi^*}\big) \xrightarrow{p} 0, \qquad \text{i.e.,} \qquad \hat{\theta}_{\mathrm{act}}^{\hat{\pi}} - \hat{\theta}_{\mathrm{act}}^{\pi^*} = o_p(n^{-1/2}).$$

Consequently, if the oracle estimator satisfies $\sqrt{n}(\hat{\theta}_{\mathrm{act}}^{\pi^*} - \theta^*) \Rightarrow \mathcal{N}(0, \Sigma_*)$, then by Slutsky's theorem,

$$\sqrt{n}(\hat{\theta}_{\mathrm{act}}^{\hat{\pi}} - \theta^*) = \sqrt{n}(\hat{\theta}_{\mathrm{act}}^{\pi^*} - \theta^*) + o_p(1) \Rightarrow \mathcal{N}(0, \Sigma_*).$$

$\square$

## B. Additional details of the algorithm and experiments

In all of our experiments, we treat the estimates obtained from the full dataset as the true value. In each trial, the underlying data points are fixed, and the randomness arises from the labeling decisions $\xi_i$. The results are averaged over 3000 trials to obtain the final estimate. In the Income and Perioperative datasets, the predictive model $\mu$ and the uncertainty function $V(x)$ are both estimated using the XGBoost regression model. And for the Political Bias dataset, the predictive model $\mu$ is based on the predictions of GPT-3.5, while the uncertainty function $V(x)$ is estimated using the predictions from GPT-4.

### B.1. Details of estimating $V(x)$

Given a small collection of queried responses $\{Y_1', \ldots, Y_{n'}'\}$, the first-order Hoeffding component $h_1(y)$ can be estimated by

$$\hat{h}_1(y) = \frac{1}{\binom{n'}{r-1}} \sum_{\mathcal{C}_{n',r}^i} h\left(y, Y_{i_2}', \ldots, Y_{i_r}'\right).$$

The estimator $\hat{h}_1^\mu(y)$ is defined analogously by replacing $Y_i$'s with their model-based predictions $\hat{Y}_i$'s. Using these estimates, we then fit a regression function $V(x) = \mathbb{E}[|\hat{h}_1(Y) - \hat{h}_1^\mu(\hat{Y})| \mid X = x]$ via suitable machine learning approaches, such as random forests. Since $Y$ is the only unknown component in $V$, we can first fit a model for $Y$ and then compute $V$, which can improve model accuracy.

To fully exploit the available labeled data while mitigating overfitting, we adopt a careful data reuse strategy. Using the same data to both train the predictive model $\mu$ and evaluate its predictions may introduce bias. In the Income and Perioperative datasets, the total numbers of labeled samples used are $m_{\mathrm{Income}} = 1000$ and $m_{\mathrm{Perioperative}} = 760$, respectively. To address the potential reuse bias, we employ a two-fold scheme, in which the queried dataset $\mathcal{D} = \{(X_k', Y_k')\}_{k=1}^m$ is divided into two folds. Specifically, one fold $\mathcal{D}_1$ is used to train the predictive model $\mu$, while the other fold $\mathcal{D}_2$ is used to generate predictions $\{\hat{Y}_i : i \in \mathcal{D}_2\}$ and to construct the estimators $\hat{h}_1$ and $\hat{h}_1^\mu$. Since both $h_1$ and $h_1^\mu$ converge at fast rates, we then refit the uncertainty function $V$ using the full dataset $\mathcal{D}$, which avoids additional data splitting while maintaining satisfactory empirical performance. If the predictive model is pre-trained, such as the LLM, the procedure can be further simplified, where the data-splitting is not needed.

### B.2. Computational Details

We provide additional details on the computational cost of the proposed active $U$-statistic. Recall that the normalized AIPW $U$-statistic in (7) can be written as

$$U_{\mathrm{AIPW}}^{\hat{\pi}, N} = \underbrace{\frac{1}{\binom{n}{r}} \sum_{\mathcal{C}_{n,r}} h\big(\hat{Y}_{i_1}, \ldots, \hat{Y}_{i_r}\big)}_{\text{first term}} + \underbrace{\frac{1}{\hat{N}} \sum_{\mathcal{C}_{n,r}} \Delta_h(i_1, \ldots, i_r) \frac{\xi_{i_1} \cdots \xi_{i_r}}{\hat{\pi}_{i_1} \cdots \hat{\pi}_{i_r}}}_{\text{second term}}.$$

Its computation consists of three components: evaluating the first plug-in term over the full unlabeled sample of size $n$, evaluating the second correction term over the queried labeled sample of size approximately $n_b$, and constructing the sampling probabilities $\hat{\pi}_i$. Under a naive implementation, the first two components have complexities $O(n^r)$ and $O(n_b^r)$, respectively, with $n_b \ll n$. Therefore, the dominant computational burden typically comes from evaluating the first plug-in term over the full unlabeled sample.

To construct $\hat{\pi}_i$, we estimate a proxy score $V(x)$ based on the estimated first-order projections $\hat{h}_1$ and $\hat{h}_1^\mu$. These quantities are computed from a small queried or historical labeled subset of size $n'$, with complexity $O((n')^{r-1})$. We then learn $V(x)$ by fitting a regression model such as XGBoost to the resulting discrepancy values. Since these steps are performed only on the small pilot set, their computational cost is modest relative to the main estimator computation.

To make this concrete, Table 1 reports the running time of each component in one simulation run under the ACS Income setup, with $n = 80{,}000$, $n_b = 2{,}500$, and $n' = 500$. Specifically, we report the time for computing $\hat{h}_1$ and $\hat{h}_1^\mu$, constructing the sampling rule, and evaluating the first and second terms of the AIPW $U$-statistic.

*Table 1.* Running time of different computational components under the ACS Income setup. The columns "fast" and "naive" correspond to the accelerated and direct implementations, respectively.

| Method | $\hat{h}_1$ and $\hat{h}_1^\mu$ (s) | Sampling rule (s) | First term fast (s) | First term naive (s) | Second term fast (s) | Second term naive (s) |
|---|---|---|---|---|---|---|
| Active | $4.9 \times 10^{-4}$ | 1.24 | $5.5 \times 10^{-3}$ | 10 | $5.1 \times 10^{-4}$ | $2.7 \times 10^{-2}$ |
| Uniform | – | – | $5.5 \times 10^{-3}$ | 10 | $4.2 \times 10^{-4}$ | $2.2 \times 10^{-2}$ |
| Classical | – | – | – | – | $3.4 \times 10^{-4}$ | $2.1 \times 10^{-2}$ |

For some special $U$-statistics, such as the Gini coefficient, fast exact algorithms are available. For example, after sorting, the computation of certain pairwise sums can be reduced from $O(n^2)$ to $O(n \log n)$. Therefore, Table 1 reports the running times under both the naive implementation and the accelerated implementation. The results show that the main computational burden comes from evaluating the first plug-in term, while the cost of constructing the sampling rule is mainly driven by regression-model fitting. In contrast, the computation of $\hat{h}_1$ and $\hat{h}_1^\mu$ is relatively inexpensive. More generally, distributed algorithms or incomplete $U$-statistics can also be incorporated to further reduce the computational cost.

### B.3. Additional figure for comparing the ratio of saved budget

We report the sample-budget saving ratio relative to the classical baseline for the experiments in Figures 1–3. Our proposed active sampling method is compared against the uniform-sampling baseline. The results show that incorporating predictions improves efficiency: uniform sampling achieves a positive saving ratio relative to the classical estimator. Moreover, our approach yields substantially larger efficiency gains than the other two benchmarks.

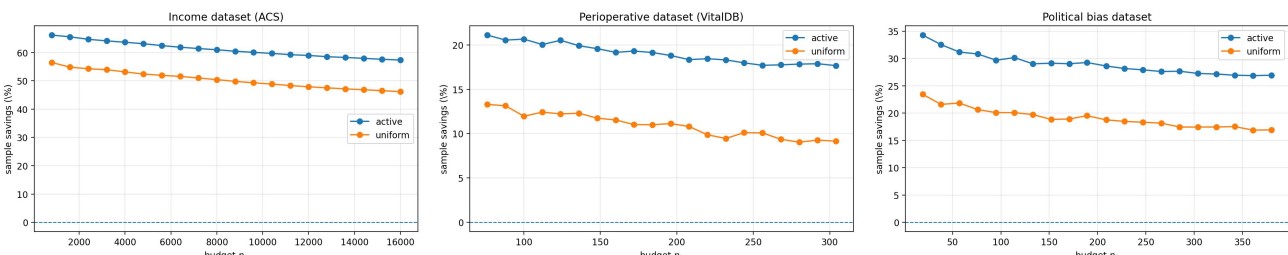

*Figure 4.* Save in sample budget due to active inference. Reduction in sample size required to achieve the same confidence interval width across the applications shown in Figures. 1–3.

## C. Additional experiment results

### C.1. Additional Experiment on a Higher-Order U-Statistic

We further consider the **UCI Bike Sharing** dataset, which contains hourly and daily records of bike rental counts together with weather and calendar covariates. Each observation corresponds to a rental record, and the label $Y$ is defined as the

log-transformed rental count,

$$Y = \log(1 + \text{cnt}).$$

The total sample size is $n = 17{,}379$, and we use $1{,}000$ initially labeled samples to construct the prediction and sampling components.

**Target parameter: Third central moment**  Beyond pairwise targets such as the Gini index or Kendall's tau, we also evaluate our method on a higher-order U-Statistic. Specifically, we consider the third central moment of the log-transformed rental count,

$$\theta^* = \mathbb{E}\big[(Y - \mathbb{E}Y)^3\big],$$

which measures the skewness structure of the rental-count distribution and can be represented as a third-order $U$-statistic. This experiment is designed to examine whether the proposed active inference framework remains effective beyond pairwise $U$-statistic targets.

Table 2 compares the proposed active sampling method with the classical and uniform baselines under different labeling budgets $n_b$. The table reports the effective sample size required by the classical estimator to match each method's confidence-interval length, with the corresponding sample savings shown in parentheses. Across all labeling budgets, the active method achieves the largest effective sample size and consistently improves over both baselines. This demonstrates that our approach remains efficient for higher-order $U$-statistic tasks.

*Table 2.* Effective sample size on the UCI Bike Sharing dataset for estimating the third central moment of the log-transformed rental count. The numbers in parentheses denote the corresponding sample savings relative to the classical baseline.

| Method | $n_b = 1892$ | $n_b = 2584$ | $n_b = 3275$ |
|---|---|---|---|
| active | 2843 (50%) | 3820 (47%) | 4795 (46%) |
| classical | 1892 | 2584 | 3275 |
| uniform | 2683 (41%) | 3593 (39%) | 4504 (37%) |

## C.2. Comparison of sampling policies

We compare our oracle sampling policy

$$\pi(X) \propto \sqrt{\mathbb{E}[|h_1(Y) - h_1^\mu(\hat{Y})|^2 \mid X]}$$

with the suboptimal one introduced by Zrnic & Candès (2024a) for M-estimator, where $\pi^Y(X) \propto \sqrt{\mathbb{E}[|Y - \hat{Y}|^2 \mid X]}$. AIPW U-statistics based on sampling probability $\pi^Y(X)$ is denoted as **plugin-act-Y**.

We generate i.i.d. covariates from a multivariate normal distribution

$$X_i \sim \mathcal{N}(0, \Sigma), \qquad i = 1, \ldots, n,$$

where

$$\Sigma = 0.3\, I_p + 0.7\, \mathbf{1}_p \mathbf{1}_p^\top,$$

The response is generated from a nonlinear regression model with Gaussian noise:

$$Y_i = \mu + f(X_i) + \varepsilon_i, \qquad \varepsilon_i \sim \mathcal{N}(0, \sigma^2),\ \sigma = 0.3,$$

where the regression function depends on the first four coordinates,

$$f(X_i) = 1.2 \sin(X_{i1}) + 0.8 \cos(X_{i2}) + 0.6\, X_{i1} X_{i2} + 0.5\, X_{i3}^2 + 0.7\, \tanh(X_{i4}).$$

Figure 5 compares empirical coverage (left) and effective sample size $n_{\text{eff}}$ (right) across budgets $n_b$. In contrast, the **plugin-act-Y** method improves over **Uniform** but remains uniformly less efficient than **Active**. This gap is consistent with the theorem 2.2: the variance-optimal design is driven by he first-order Hoeffding projection, $|h_1(Y) - h_1^\mu(\hat{Y})|$, rather than the mean-estimation residual $|\hat{Y} - Y|$ (the two coincide only in the special case $r = 1$ and $h(y) = y$).

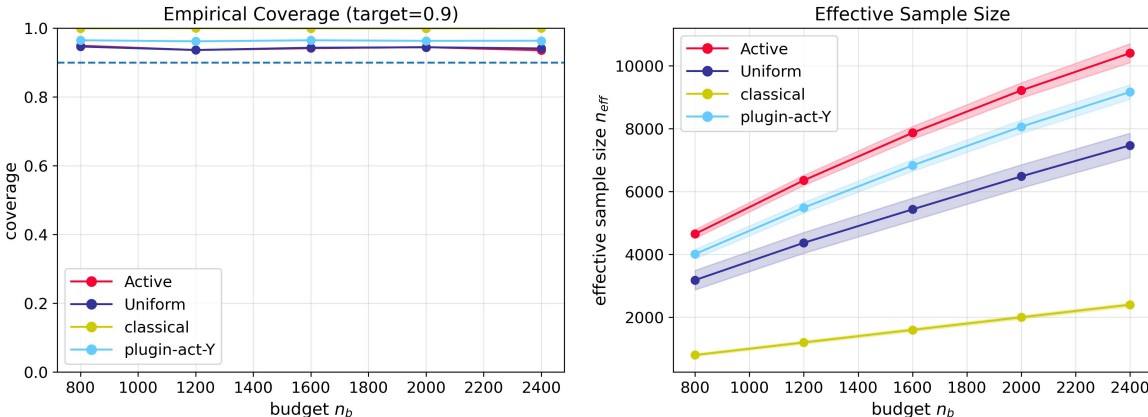

*Figure 5.* Estimation of Gini index. **Left**: empirical coverage of the intervals. **right**: effective sample size $n_{\text{eff}}$, with shaded $\pm 1$ standard-deviation bands.

### C.3. Sensitivity to Predictive-Model Misspecification

We further examine the sensitivity of the proposed method to predictive-model misspecification. The validity of the AIPW $U$-statistic does not require the predictive model $\mu$ to be correctly specified, since the plug-in term is corrected by the IPW label term. Thus, misspecification mainly affects efficiency through the quality of the proxy score used to construct the sampling rule, rather than the validity of the inference procedure.

To assess this issue, we conduct an additional experiment on the perioperative dataset. In this experiment, the model used to predict the outcome is intentionally misspecified by using a linear model. Table 3 reports the effective sample size under different labeling budgets $n_b$. The numbers in parentheses denote the corresponding sample savings relative to the classical baseline.

*Table 3.* Effective sample size on the perioperative dataset under predictive-model misspecification. The prediction model is intentionally misspecified by using a linear model. The numbers in parentheses denote the corresponding sample savings relative to the classical baseline.

| Method | $n_b = 76$ | $n_b = 208$ | $n_b = 304$ |
|---|---|---|---|
| active | 85 (11%) | 241 (15%) | 362 (19%) |
| classical | 76 | 208 | 304 |
| uniform | 68 (-11%) | 190 (-8%) | 283 (-6%) |

The results show that the proposed active method is reasonably robust to predictive-model misspecification. In this setting, the uniform AIPW estimator can be less efficient than the classical estimator, as indicated by the negative sample savings. In contrast, the active method consistently achieves larger effective sample sizes across all labeling budgets. This suggests that the efficiency gain of the proposed method is not solely driven by accurate prediction itself, but also by whether the estimated sampling probabilities successfully prioritize informative samples.

### C.4. Illustration of the variance reduction through normalization

We compare the original AIPW $U$-statistic $U_{\text{AIPW}}^{\pi}$ with its Hájek-normalized variant $U_{\text{AIPW}}^{\pi,N}$ in (7). We consider the two additional benchmarks:

- **uniform-prime**: AIPW $U$-statistic in (3) with uniform sampling policy without normalization, i.e. $\pi(x) \equiv n_b/n$.

- **active-prime**: Our AIPW $U$-statistic without normalization.

Figure 6 compares the Hájek normalized and unnormalized AIPW U-statistics. Across the considered labeling budgets, the normalized estimator exhibits smaller MSE in this experiment.

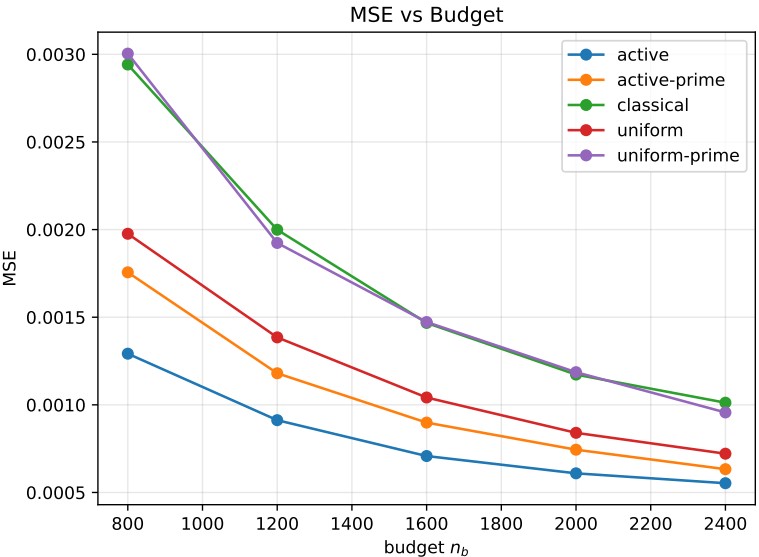

*Figure 6.* Average MSE under different choices of sample budget $n_b$.

## C.5. $U$-estimator

We conduct a simulation study to evaluate the performance of the proposed active $U$-estimator compared with two baseline methods under varying data conditions:

- **noML:** uses only the labeled subset $(X, Y)$ selected by uniform sampling. The parameter $\theta$ is obtained by minimizing $L_{n_b}(\theta)$ with the pairwise logistic loss.

- **Semi:** pseudo-labels are generated for unlabeled data by a pre-trained regression model $f_{\text{pseudo}}(X)$, trained on $(X_1, Y_1)$, and the loss is computed using both labeled and pseudo-labeled pairs:

$$\tilde{L}(\theta) = \frac{1}{n_b(n_b - 1)} \sum_{i \neq j} \ell(\theta; Z_i, \hat{Z}_j), \quad \hat{Z}_j = (X_j, \hat{Y}_j).$$

- **Act:** our active $U$-estimator based on the two-stage procedure in Section 3.1.

We assume that the data are generated from a linear model

$$Y = X^\top \theta^* + \varepsilon, \qquad \varepsilon \sim \mathcal{N}(0, 1),$$

where $X \in \mathbb{R}^p$ follows a multivariate Gaussian distribution $\mathcal{N}(0, I_p)$, and $\theta^*$ is the ground-truth coefficient vector normalized to $\|\theta^*\|_2 = 1$. The goal is to estimate $\theta^*$ based on pairwise comparisons of $(X_i, Y_i)$ using $U$-statistic–type losses.

The pairwise logistic loss is defined as

$$\ell(\theta; Z_i, Z_j) = \log\big(1 + \exp\big[-s_{ij}\, \theta^\top (X_i - X_j)\big]\big), \qquad s_{ij} = \text{sign}(Y_i - Y_j),$$

and the empirical $U$-risk is given by

$$L_n(\theta) = \frac{1}{n(n - 1)} \sum_{i \neq j} \ell(\theta; Z_i, Z_j).$$

We observe an unlabeled set $(X_{\text{unl}}, Y_{\text{unl}})$ of size $1,500$, from which a small subset is actively queried. A historical labeled set $(X_{lab}, Y_{lab})$ with same size is used for supervised training, compute the pilot estimator and pseudo-labeling or gradient reference. We compare the proposed active sampling strategy with the classical and uniform baselines in Figure 7. Over the entire range of budgets, active U-estimation yields a larger effective sample size and a lower MSE than either baseline, reflecting a clear improvement in statistical efficiency.

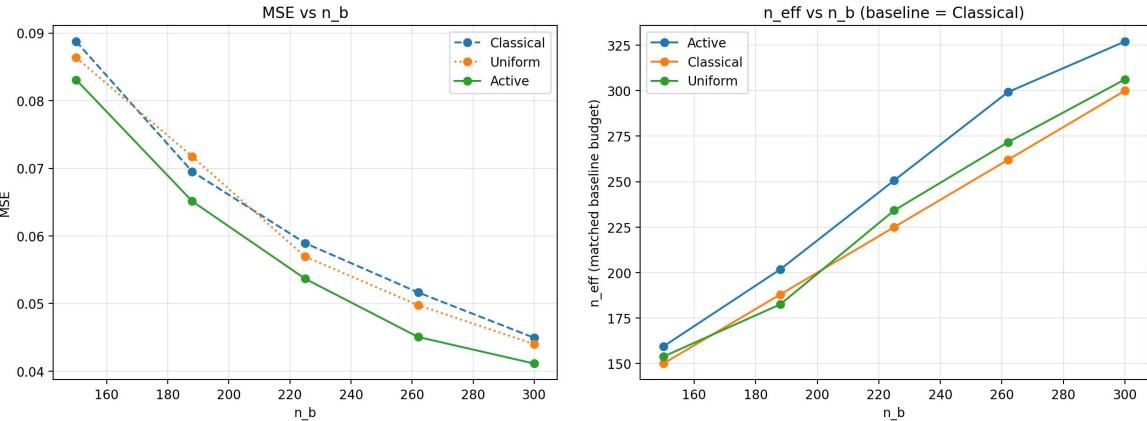

*Figure 7.* **Left**: Average MSE for the U-estimation under different choices of sample budget $n_b$. **Right**: $n_{\text{eff}}$ for the U-estimator under different choices of sample budget $n_b$.

## D. Discussion on Degenerate $U$-Statistics

Our main theory focuses on the non-degenerate regime, in which the first-order Hoeffding projection determines the leading asymptotic behavior. In this setting, the optimal sampling rule is driven by the residual uncertainty in the first-order projection, as established in Theorem 2.2.

When the target $U$-statistic is degenerate, however, the first-order projection vanishes and the leading contribution comes from higher-order Hoeffding components. The resulting limiting distribution of the AIPW $U$-statistic is generally non-Gaussian, which makes the construction of valid confidence intervals substantially more delicate. Nevertheless, the AIPW $U$-statistic itself remains well defined in the degenerate case. Thus, the proposed framework can still be applied with a generic sampling policy, such as uniform sampling. The main difficulty lies instead in deriving a variance-optimal sampling policy and establishing the corresponding inferential theory.

Below, we outline a possible route toward deriving an optimal sampling policy for second-order degenerate $U$-statistics. Define

$$\Phi_2(y_1, y_2) := \mathbb{E}[h(y_1, y_2, Y_3, \ldots, Y_r)], \qquad \Phi_2^\mu(\hat{y}_1, \hat{y}_2) := \mathbb{E}\Big[h(\hat{y}_1, \hat{y}_2, \hat{Y}_3, \ldots, \hat{Y}_r)\Big],$$

and let

$$\Delta_2 := \Phi_2(Y_1, Y_2) - \Phi_2^\mu(\hat{Y}_1, \hat{Y}_2).$$

In this regime, the leading variance term depending on the sampling policy $\pi(\cdot)$ is of the form

$$\mathbb{E}\left[\Delta_2^2 \left\{ \frac{1}{\pi(X_1)\pi(X_2)} - 1 \right\}\right].$$

Thus, up to terms independent of $\pi$, the oracle policy can be characterized by

$$\min_\pi \mathbb{E}\left[\frac{\Delta_2^2}{\pi(X_1)\pi(X_2)}\right] \quad \text{s.t.} \quad \mathbb{E}\{\pi(X)\} \le \frac{n_b}{n}, \quad 0 < \pi(X) \le 1.$$

Unlike the non-degenerate case, this criterion depends on a pairwise quantity rather than a one-point score, and therefore does not generally reduce to a closed-form marginal rule. Writing

$$T_2(x, x') := \mathbb{E}\big[\Delta_2^2 \mid X_1 = x, X_2 = x'\big],$$

the corresponding optimality condition suggests a fixed-point relation of the form

$$\pi^*(x) \propto \sqrt{\mathbb{E}\left[\frac{T_2(x, X')}{\pi^*(X')}\right]},$$

where $X'$ is an independent copy of $X$. This expression shows that a closed-form solution is generally unavailable. A possible practical approach is to estimate the pairwise score $T_2(x, x')$ from a pilot labeled sample and then solve the resulting policy by a fixed-point or alternating optimization procedure. This suggests that extending active inference from non-degenerate to degenerate $U$-statistics requires fundamentally new tools for both policy design and asymptotic inference.

