# OpenReview forum: "Learning $U$-Statistics with Active Inference"
_ICML.cc/2026/Conference — ICML 2026 regular_

### Official Review · Reviewer_GsBd · 2026-02-26

**Soundness:** 4
**Presentation:** 3
**Significance:** 4
**Originality:** 4
**Overall Recommendation:** 6
**Confidence:** 4

**Summary:**

This paper introduces an active inference framework specifically designed for U-statistics, addressing the challenge of costly label acquisition in statistical learning. While existing active inference methods primarily target mean estimation or convex M-estimation, they are not directly applicable to U-statistics due to their complex combinatorial structure and the dependence between sample tuples. To resolve this, the authors propose an Augmented Inverse Probability Weighting (AIPW) U-statistic that effectively integrates machine learning predictions with adaptively queried labels to improve estimation efficiency.

**Compliance With Llm Reviewing Policy:**

Affirmed.

**Final Justification:**

The work presents a solid theoretical contribution to active sampling with U-statistics, supported by convincing experimental evidence. Given the technical rigor and the significance of the results, I have increased my score to Strong Accept. I highly recommend this paper.

**Key Questions For Authors:**

.

**Limitations:**

Yes

**Strengths And Weaknesses:**

The submission is technically sound and provides a rigorous extension of active statistical inference to the domain of U-statistics. The authors effectively address the combinatorial challenges inherent in U-statistics by developing an Augmented Inverse Probability Weighting (AIPW) framework. A major strength in soundness is the characterization of the optimal sampling rule, which is theoretically proven to minimize asymptotic variance by focusing on the residual uncertainty of the first-order Hoeffding projection rather than simple prediction residuals. This theoretical claim is well-supported by a coupling-based proof for asymptotic normality and consistent empirical results across multiple real-world datasets, where the method achieves nominal coverage while significantly reducing the required labeling budget. In terms of significance, the paper addresses a highly relevant problem in modern machine learning: performing valid statistical inference when labels are costly and budgets are limited. By enabling more efficient estimation of Interaction-based metrics like the Gini index, AUC, and Kendall's tau, the work has broad potential impact in fields ranging from socioeconomic analysis to medical diagnostics. The extension to U-statistic-based empirical risk minimization further broadens its significance, as it provides a robust toolkit for label-efficient ranking and metric learning.

The work demonstrates strong originality by bridging the gap between recent advances in prediction-powered inference and the classical theory of U-statistics. While prior work has focused on simpler M-estimation, this paper introduces novel algorithmic and analytical techniques to handle the dependence across sample tuples. The adaptation of Hájek normalization to the U-statistic setting is a creative combination of survey sampling principles and modern active learning that successfully mitigates finite-sample variance inflation. The presentation is clear, well-structured, and easy to follow. The authors provide a helpful narrative that motivates the need for active inference before delving into the formal mathematical framework. The submission does an excellent job of positioning itself within existing literature, clearly explaining how it generalizes recent mean-estimation frameworks to more complex kernels. The inclusion of practical sampling strategies and detailed experimental results, including effective sample size metrics, provides sufficient information for researchers to understand and potentially reproduce the results.

A minor weakness is that the current theoretical analysis and sampling policies are primarily focused on non-degenerate U-statistics. While the authors acknowledge this and suggest that identifying optimal rules for the degenerate regime remains an open problem, a more detailed discussion of the framework's performance or potential heuristics in such cases would have further strengthened the paper. Additionally, the computational cost of constructing augmented estimators from all unlabeled samples could be demanding in very large-scale settings. However, these points do not significantly detract from the overall high quality and contribution of the work.

---

> ### Author Rebuttal · Authors · 2026-03-30
>
> We thank the reviewer for this constructive comment and for recognizing that these issues do not detract from the main contribution of the paper.
>
> **To W1**:  We agree that the current theory is focused on the non-degenerate regime; extending optimal active sampling to the degenerate case is nontrivial because **higher-order Hoeffding components** become dominant, although the AIPW construction itself remains **applicable under suitable sampling policies**.  Owing to space constraints, the details are provided in our Response to **Reviewer TFTP, Q3**.
>
> ***
>
> **To W2**: We also agree that computational cost deserves clearer discussion.  We now discuss this issue in more detail below and will incorporate the same clarification into the revision. To better explain the computational structure of the proposed method, recall that our AIPW U-statistics is
>
> $U_{\rm act} = U_{\rm plug} + U_{\rm corr},$
>
> where
> $U_{\rm plug}=\binom{n}{r}^{-1}\sum_{(i_1,\ldots,i_r)\in\mathcal C_{n,r}}h(\hat Y_{i_1},\ldots,\hat Y_{i_r}),$
> and $U_{\rm corr}=\binom{n}{r}^{-1}\sum_{(i_1,\ldots,i_r)\in\mathcal C_{n,r}}\Delta_h(i_1,\ldots,i_r)\prod_{j=1}^r \frac{\xi_{i_j}}{\hat\pi_{i_j}}.$
>
> Its computation has three components:
> 1. evaluating the first term (plug) over the full unlabeled sample of size$ n $;
> 2. evaluating the second term (corr) over the queried labeled sample of size $ n_b $ (the second term);
> 3. constructing the sampling probabilities $ \hat\pi $.
>
> The first two components have complexities $ O(n^r) $ and $ O(n_b^r) $ respectively, with $ n_b\ll n $. To construct $ \hat\pi $, we estimate a proxy score $ V(x) $ based on the estimated first-order projections $ \hat h_1 $ and $ \hat h_1^\mu $.  These quantities are computed from a small queried or historical labeled subset of size $ n' $,   with complexity $ O(n'^{r-1}) $.  We then learn $ V(x) $ by fitting a regression model such as XGBoost to the resulting discrepancy values.  Because these steps are performed only on the small pilot set, their computational cost is negligible compared with the main estimator computation. Consequently, the dominant computational burden remains the evaluation of the first term of the AIPW U-statistic over the full unlabeled sample, with complexity $ O(n^r) $.
>
>  To make this concrete, we report in the table below the running time of each component in one simulation run under the ACSIncome setup, with  $ n=80,000 $, $ n_b=2,500 $ and $ n'=500 $.   Specifically, we report the time for computing $ h_1 $ and $ h_1^\mu $,  constructing the sampling rule, and evaluating the first and second terms of the AIPW U-statistic.
>
> | | $ h_1 $ and $ h_1^{\mu} $ (s) | sampling_rule (s) | first_term_fast (s) | first_term_naive (s) | second_term_fast (s) | second_term_naive (s) |
> | --- | --- | --- | --- | --- | --- | --- |
> | active | $ 4.9*10^{-4} $ | 1.24 | $ 5.5*10^{-3} $ | 10 | $ 5.1*10^{-4} $ | $ 2.7*10^{-2} $ |
> | uniform | \ | \ | $ 5.5*10^{-3} $ | 10 | $ 4.2*10^{-4} $ | $ 2.2*10^{-2} $ |
> | classical | \ | \ | \ | \ | $ 3.4*10^{-4} $ | $ 2.1*10^{-2} $ |
>
> For some special U-statistics, such as the Gini coefficient, fast exact algorithms are available. For example, after sorting, the computation of certain pairwise sums can be reduced from $ O(n^2) $ to $ O(n \log n) $. Therefore, the table also reports the running times under both the naive implementation and the accelerated implementation.  The results show that the main computational burden comes from evaluating the first term, while the cost of constructing the sampling rule is driven primarily by regression-model fitting.   In contrast, the computation of $ h_1 $ and $ h_1^\mu $ is relatively inexpensive.  More generally, standard approaches such as distributed algorithms or incomplete U-statistics can also be incorporated to further reduce the computational cost.

---

> > ### Author Rebuttal · Reviewer_GsBd · 2026-03-31
> >
> > Thank you for addressing all of my concerns. Everything has been resolved to my satisfaction. I appreciate your time and effort in working through this thoroughly. I have changed it to strong accept.

---

> > > ### Author Response · Authors · 2026-04-01
> > >
> > > Thank you for your thoughtful review and for your positive feedback on our revision. We sincerely appreciate your time, effort, and support, and we are especially grateful for your change of recommendation to strong accept.

---

### Official Review · Reviewer_TFTP · 2026-03-08

**Soundness:** 4
**Presentation:** 3
**Significance:** 3
**Originality:** 4
**Overall Recommendation:** 5
**Confidence:** 5

**Summary:**

This paper studies a very interesting problem of estimating U-statistics when labels are costly and only a limited labeling budget is available. The authors propose an active inference framework that combines predictive models with adaptive label acquisition. The method is built around an augmented inverse probability weighted (AIPW) U-statistic estimator that uses predictions as a plug-in estimate and corrects bias through inverse-probability weighting of queried labels. The paper derives the optimal sampling rule that minimizes the asymptotic variance of the estimator using the Hoeffding decomposition of U-statistics. The authors also provide asymptotic normality results and a consistent variance estimator, enabling valid confidence intervals under adaptive sampling. The framework is further extended to U-statistic–based empirical risk minimization problems such as ranking. Experiments on several real datasets show improved label efficiency compared to classical and uniform sampling approaches.

**Compliance With Llm Reviewing Policy:**

Affirmed.

**Key Questions For Authors:**

1. How sensitive is the sampling policy to misspecification of the predictive model used to estimate the projection quantities?

2. What is the computational complexity of the method when applied to large datasets where enumerating all U-statistic tuples is expensive?

3. Could the framework be extended to degenerate U-statistics, where the first-order projection vanishes?

**Limitations:**

yes

**Strengths And Weaknesses:**

Strength: The paper addresses a relevant problem: performing valid statistical inference when labels are expensive but unlabeled data and predictive models are available. The combination of active inference and U-statistics appears novel, and the derivation of the optimal sampling rule based on the first-order Hoeffding projection is an interesting insight. The theoretical development is reasonably complete, including asymptotic normality and variance estimation under adaptive sampling. The extension to U-statistic–based ERM broadens the applicability of the framework. Empirical results on several datasets demonstrate consistent improvements in effective sample size while maintaining coverage.

Weakness: Some aspects of the practical implementation could be clearer. In particular, estimating the projection quantities required for the optimal sampling rule may be challenging in practice, especially when the kernel or predictive model is complex. The computational cost of evaluating U-statistic terms over large datasets is also not discussed in detail. Additionally, the empirical evaluation focuses on a small number of datasets, and it would be helpful to see experiments on larger-scale problems or additional U-statistic tasks.

---

> ### Author Rebuttal · Authors · 2026-03-30
>
> **To W**:  We added a real-data experiment on the UCI Bike Sharing dataset, using the third central moment of the log-transformed rental count as the target, where $ Y=\log(1+\mathrm{cnt}) $ and $ \theta=\mathbb{E}\big[(Y-\mathbb{E}Y)^3\big]. $ The total sample size is 17,379 , and we use 1,000 initially labeled samples.  The table below reports the sample size the classical estimator needs to match each method's confidence-interval length, with parentheses showing the corresponding sample savings.  Our method still yields shorter confidence intervals than uniform sampling and clearly outperforms the classical baseline on this higher-order U-statistic task. Due to time constraints, we add one additional dataset in this revision and leave broader evaluation to future revision.
>
> | | sample size($ n_b= $1892) | sample size($ n_b= $2584) | sample size($ n_b= $3275) |
> | --- | --- | --- | --- |
> | active | 2843 (50%) | 3820 (47%) | 4795 (46%) |
> | classical | 1892 | 2584 | 3275 |
> | uniform | 2683 (41%) | 3593 (39%) | 4504 (37%) |
>
> ***
>
> **To Q1**: We thank the reviewer for raising this important question. **Our method does not require the predictive model to be correctly specified for validity**: the AIPW U-statistic remains unbiased because the prediction term is corrected by the IPW label term. In this sense, **misspecification mainly affects efficiency**, through the quality of the proxy used to construct the sampling rule, rather than correctness of the inference itself.
>
> To assess this sensitivity, we conducted an experiment on the perioperative dataset, where the model used to predict Y is intentionally misspecified by using a linear model.   We summarize the results in the table below, with the same interpretation as the previous table.  In this case, uniform AIPW may even be less efficient than the classical estimator. In contrast, our active method consistently requires fewer samples across all labeling budgets.
>
> |  | sample size($ n_b= $76) | sample size($ n_b= $208) | sample size($ n_b= $304) |
> | --- | --- | --- | --- |
> | active | 85 (11%) | 241 (15%) | 362 (19%) |
> | classical | 76 | 208 | 304 |
> | uniform | 68 (-11%) | 190 (-8%) | 283 (-6%) |
>
> These results suggest that **our method is reasonably robust to predictive-model misspecification**. This highlights that the performance improvement is driven less by perfect prediction itself and more by whether the estimated sampling probabilities successfully prioritize informative samples.
>
> ***
>
> **To Q2:**  We thank the reviewer for highlighting this important practical issue. We have included a more detailed discussion of the computational complexity in our response to **Reviewer GsBd, W2**, and we will make this discussion more explicit in the revision.
>
> ***
>
> **To Q3:**  Our current theory is restricted to the non-degenerate regime, since the optimal sampling rule in the paper is derived from the first-order Hoeffding projection. In addition, under degeneracy, the limiting distribution of AIPW U-statistics is generally non-normal, making the construction of valid confidence intervals for the parameter of interest nontrivial. Nevertheless, **the AIPW U-statistic itself remains well defined for degenerate U-statistics, so the framework can still be used with a suitable  sampling policy**. What becomes substantially more difficult is **the derivation of an optimal policy and the corresponding theory**.
>
> We provide a possible solution to derive the optimal sampling policy under a second-order degenerate case. Define
>
>  $ \Phi_2(y_1,y_2):=\mathbb{E}\left[h(y_1,y_2,Y_3,\dots,Y_r)\right],\qquad\Phi_2^\mu(\hat y_1,\hat y_2):=\mathbb{E}\left[h(\hat y_1,\hat y_2,\hat Y_3,\dots,\hat Y_r)\right],\qquad \Delta_2:=\Phi_2(Y_1,Y_2)-\Phi_2^\mu(\hat Y_1,\hat Y_2).$
>
> In this regime, the leading variance term that depends on the sampling policy $ \pi(\cdot) $ is
> $ \mathbb{E}\left[\Delta_2^2\left(\frac{1}{\pi(X_1)\pi(X_2)}-1\right)\right]. $ Hence the oracle policy is characterized by
>
> $ \min_{\pi}\;\mathbb{E}\left[\frac{\Delta_2^2}{\pi(X_1)\pi(X_2)}\right]\quad\text{s.t. }\mathbb E[\pi(X)]\le \frac{n_b}{n}, 0<\pi(X)\le1. $
>
> Unlike the non-degenerate case, this criterion depends on a **pairwise** quantity rather than a one-point score, and thus does not generally reduce to a closed-form marginal rule. Writing
>
> $ T_2(x,x'):=\mathbb{E}\left[\Delta_2^2\mid X_1=x,X_2=x'\right], $
>
>  the corresponding optimality condition suggests a fixed-point relation of the form
>
> $ \pi(x) \propto \sqrt{\mathbb{E}\left[\frac{T_2(x,X')}{\pi(X')}\right]}$
>
> This shows that, although a closed-form solution is generally unavailable, the problem may still be approached through **pilot estimation** of  $ T_2(x,x')$  followed by a **fixed-point or alternating optimization procedure**.  We will incorporate the above discussion in the revision and highlight it as an important direction for future research.

---

> > ### Author Rebuttal · Reviewer_TFTP · 2026-04-02
> >
> > My main concerns were addressed and added clarification was also provided. Thanks for the detailed response, I am keeping my score.

---

> > > ### Author Response · Authors · 2026-04-03
> > >
> > > Thank you for your thoughtful review and for taking the time to read our response carefully.  We sincerely appreciate your time, feedback, and consideration.

---

### Official Review · Reviewer_HUKd · 2026-03-10

**Soundness:** 2
**Presentation:** 3
**Significance:** 2
**Originality:** 3
**Overall Recommendation:** 4
**Confidence:** 3

**Summary:**

This paper studies the active inference framework for U-statistics. Roughly speaking, the goal of the problem is to estimate the quantity $\theta^* = E[h(Y_1,\dots,Y_r)]$, using iid examples $Y_1,\dots,Y_n$. In many cases, one gets data $(X_1,Y_1),\dots,(X_n,Y_n)$, but only observe the covariate part $X_i$ and need to query $Y_i$ to reduce the cost of estimation.

This paper studies augmented inverse probability weighting U statistics, where one can get some ML predicted $\hat{Y}$ for $X$. Using such a prediction, the paper proposes a sampling strategy $\pi(X)$, the probability of requesting the label $Y$ to minimize the variance of
augmented inverse probability weighting U statistics, when the expected number of queries is fixed. Furthermore, the paper extends the framework and analysis to the case of empirical risk minimization and conducts experiments on real datasets to show the performance of the proposed method.

**Compliance With Llm Reviewing Policy:**

Affirmed.

**Final Justification:**

After rebuttal, I will keep my positive score for the paper

**Key Questions For Authors:**

1. Can you discuss how to estimate the quantity $s(X), V(X)$? Would estimating these quantities be costly?

2. Can you explain how the ML predictions are involved in the experiments? Also, can you explain in theory how the accuracy of the ML prediction affects the performance of the proposed method? What if we do not have the prediction?

3. If possible, can you make a technical comparison between this work and prior works?

**Limitations:**

Yes

**Strengths And Weaknesses:**

Strength:

The setting studied by this paper is well-motivated by practical applications. In particular, this paper takes the ML prediction into consideration, which also broadly captures recent interests from a broader ML community. Furthermore, this paper not only derives rigorous theoretical guarantees for the proposed method, but also shows the proposed method is useful over real datasets

Weakness:

I think some of the details of the paper are not discussed thoroughly. For example, the proposed method asks to estimate quantities such as $s(X)$ or $V(X)$, but the paper does not discuss how these quantities are estimated. It seems to me that these quantities need to be estimated for the covariate $X$, which needs to implement an $r-1$ order U-statistic, and are still costly. Another example is in the experimental part, the paper does not specify how the ML prediction is used for the income dataset or perioperative dataset, so it is not clear to me what the precise setting of the experiments is.

Another minor weakness is that this paper does not present a clear finite sample analysis, so it is unclear from the theory how much the proposed method can benefit us.

---

> ### Author Rebuttal · Authors · 2026-03-30
>
> # (1) How are $V(X)$ and $s(X)$ estimated, and is this costly?
>
> Our oracle rule depends on
>
> $s(X)=\mathbb{E}[|h_1(Y)-h_1^\mu(\hat Y)|^2\mid X ].$
>
> In practice, we use a learned proxy
>
> $V(x)=\mathbb{E}[|\hat{h}_1(Y)-\hat{h}^\mu_1(\hat{Y})|\mid X=x],$
>
> to approximate it. Here $\hat{h}_1$ and $\hat{h}^\mu_1$ are estimated from a small queried or historical labeled subset of size $n'$, with complexity $O(n'^{r-1})$. We then learn $V(x)$ by fitting a regression model such as XGBoost to the resulting discrepancy values. Since this is done only on the pilot set, the extra cost is modest relative to the main estimator. These details are in Appendix B.1, and we will clarify them in the main text. Distributed algorithms or incomplete U-statistics can further reduce computation.
>
> We also verified this discussion empirically; due to space limits, the results are included in **our response to Reviewer GsBd, W2**.
>
> # (2) How are the ML predictions and related explanations?
>
> The prediction model $\mu$ is used in two ways: generate pseudo-labels $\hat Y=\mu(X)$ for the plug-in part of the AIPW U-statistic, and help construct the uncertainty score $V(x)$ for active sampling.
>
> For income dataset, X consists of demographic covariates and $\mu(X)$ predicts income label Y, which is used in the AIPW estimator for the Gini index. For VitalDB dataset, we construct $D=Y^a-Y^b$ pre/post measurements and predict $D$ from case-level covariates for the Wilcoxon signed-rank target. In both datasets, $\mu$ and $V(x)$ are implemented using XGBoost.  These details are in Appendix B.1, and we will state them more clearly in the main text.
>
> Importantly, **validity does not require **$\mu$** to be perfectly specified; **$\mu$** is mainly to improve efficiency**. Theorems 2.2 and 2.4 imply that the asymptotic variance of the proposed active estimator is
>
> $\sigma_{\rm act}^2=r^2\left[\operatorname{Var}(h_1(Y))+\frac{n}{n_b}\bigl(\mathbb{E}[\sqrt{s(X)}]\bigr)^2-\mathbb{E}[s(X)]\right].$
>
> The corresponding variances of the benchmark methods are
>
> $\sigma_{\rm unif}^2=r^2\left[\operatorname{Var}(h_1(Y))+\frac{n}{n_b}\mathbb{E}[s(X)]-\mathbb{E}[s(X)]\right].$
>
> $\sigma_{\rm classical}^2=r^2\left[\operatorname{Var}(h_1(Y))+\frac{n}{n_b}\mathbb{E}[h_1(Y)^2]-\mathbb{E}[h_1(Y)^2]\right].$
>
> So a better predictive model $\mu $ reduces the projection discrepancy $\hat{h}_1(Y)-\hat{h}_1^\mu(\hat{Y})$, and hence makes $\mathbb{E}[s(X)]\ll\mathbb{E}[h_1(Y)^2]$.  As a result, the uniform estimator can achieve a substantial efficiency gain relative to the classical estimator. Moreover, since $\bigl(\mathbb{E}[\sqrt{s(X)}]\bigr)^2\leq\mathbb{E}[s(X)]$ by Jensen's inequality, the active estimator is always no worse than the uniform AIPW baseline. Intuitively, active sampling allocates more labels to points where $\mu$ is less accurate in this projection-based sense, so even an imperfect $\mu$ can improve efficiency when $s(X)$ captures where uncertainty is concentrated.
>
> If no prediction model is available, the framework remains valid but yields smaller efficiency gains; one can then use classical IPW with uniform sampling, or label a small pilot set to train a preliminary predictor before applying active sampling.
>
> # (3) Technical comparison and finite sample analysis.
>
> Compared with prior active inference work for M-estimation targets, our setting involves U-statistics with tuple dependence.  Accordingly,   the usual residual-based rule is no longer variance-optimal. Theorem 2.2 shows that the optimal policy is instead driven by the first-order Hoeffding projection residual, which is specific to U-statistic.
>
> On the theory side, we further develop a coupling-based proof route for asymptotic normality of estimator from learned-policy. Prior work such as Zrnic and Candès (2024) considers a scalar tuning family $\pi_\eta(x)=\eta u(x)$ and transfers validity from the oracle policy to the learned one through an eventual-equality argument  $P(\hat{\eta}\neq\eta^*)\to0$.
>
> In contrast, our argument is formulated directly for general plug-in sampling policies. Specifically, we use the CRN technique which places the learned policy $\hat\pi$ and the oracle policy $\pi^* $ on the same probability space. This yields a reusable comparison lemma for the AIPW terms, allowing us to weaken the eventual-equality condition by uniform policy consistency $\|\pi^*-\hat\pi\|\to0$.
>
> For finite sample analysis, we clarify that statistical efficiency is our primary objective for inference, so we adopt the asymptotic variance as the measurement. Moreover, our method already includes practical devices for finite-sample stability: Hájek-type normalization and trimming reduce variance inflation from very small sampling probabilities. Appendix C.2 shows that normalization improves stability and lowers MSE. We agree that formal finite-sample guarantees are important and can help understand the proposed method, and we will include this point in the discussion section.

---

> > ### Author Rebuttal · Reviewer_HUKd · 2026-03-31
> >
> > Thanks to the authors for their response. I will keep my score.

---

> > > ### Author Response · Authors · 2026-04-01
> > >
> > > Thank you very much for your thoughtful review and for your follow-up. We are glad that our response has adequately addressed your concerns, and we sincerely appreciate your time and consideration.

---

### Decision · Program_Chairs · 2026-04-30

**Decision:**

Accept (regular)

**Comment:**

The paper proposes an augmented IPW method to selectively query samples to enable valid statistical inference based on learned models. The contribution characterizes optimal sampling rule and demonstrates utility for ERM with some empirical evaluation. All reviewers are in consensus that the problem is well studied and the contribution well thought through. There were some clarification questions regarding specific quantities, questions regarding senstivity to a mispecified ML predictive model, extension to degenerate U-statistics, and computational complexity. Almost all reviewers' concerns were addressed. Given the consensus of all reviewers, I recommend an accept and encourage the authors to incorporate all reviewer comments, including the specific clarifications in the final version of the paper.